# Activation of astrocytes in hippocampus decreases fear memory through adenosine A₁ receptors

Yulan Li[1,2†], Lixuan Li[1,2†], Jintao Wu[1,2], Zhenggang Zhu[1,2], Xiang Feng[1,2], Liming Qin[1,2], Yuwei Zhu[1,2], Li Sun[1,2], Yijun Liu[1,2], Zilong Qiu[3], Shumin Duan[1,2,4*], Yan-Qin Yu[1,2*]

[1]Department of Neurobiology and Department of Neurology of Second Affiliated Hospital, Zhejiang University School of Medicine, Hangzhou, China; [2]NHC and CAMS Key Laboratory of Medical Neurobiology, MOE Frontier Science Center for Brain Research and Brain-Machine Integration, School of Brain Science and Brain Medicine, Zhejiang University, Hangzhou, China; [3]Institute of Neuroscience, State Key Laboratory of Neuroscience, Chinese Academy of Sciences, Shanghai, China; [4]Research Units for Emotion and Emotion Disorders, Chinese Academy of Medical Sciences, Hangzhou, China

**Abstract** Astrocytes respond to and regulate neuronal activity, yet their role in mammalian behavior remains incompletely understood. Especially unclear is whether, and if so how, astrocyte activity regulates contextual fear memory, the dysregulation of which leads to pathological fear-related disorders. We generated *GFAP-ChR2-EYFP* rats to allow the specific activation of astrocytes in vivo by optogenetics. We found that after memory acquisition within a temporal window, astrocyte activation disrupted memory consolidation and persistently decreased contextual but not cued fear memory accompanied by reduced fear-related anxiety behavior. In vivo microdialysis experiments showed astrocyte photoactivation increased extracellular ATP and adenosine concentrations. Intracerebral blockade of adenosine A₁ receptors (A₁Rs) reversed the attenuation of fear memory. Furthermore, intracerebral or intraperitoneal injection of A₁R agonist mimicked the effects of astrocyte activation. Therefore, our findings provide a deeper understanding of the astrocyte-mediated regulation of fear memory and suggest a new and important therapeutic strategy against pathological fear-related disorders.

**\*For correspondence:**
duanshumin@zju.edu.cn (SD);
yanqinyu@zju.edu.cn (Y-QY)

†These authors contributed equally to this work

**Competing interests:** The authors declare that no competing interests exist.

## Introduction

Astrocytes are the most abundant glial cells in the central nervous system (*Allen and Barres, 2009*; *Bushong et al., 2002*) and are recognized for their classical supportive, metabolic, and protective roles (*Allaman et al., 2011*; *Oliveira et al., 2015*). In addition, increasing evidence has shown that they are actively involved in modulating synaptic transmission and plasticity (*Allen and Eroglu, 2017*; *Chen et al., 2019*; *Weiss et al., 2019*; *Zhang et al., 2003*). Astrocytes respond to neuronal activity with a transient increase in the cytosolic Ca²⁺ concentration, as a result triggering the release of gliotransmitters, which in turn causes feedback regulation of neuronal activity and synaptic transmission (*Araque et al., 2014*; *Zhang et al., 2003*). However, most of these studies were performed at the synaptic or cellular level, and the roles of astrocytes in mammalian behavior remain incompletely understood.

Memory is the biological process of retaining and retrieving what we learn over time, and it is crucial for survival. However, remembering traumatic fearful events can be maladaptive, leading to both inappropriate behavioral responses and grave physical or psychological harm (*Izquierdo et al.,*

**eLife digest** Memory is the record of what we learn over time and is essential to our survival. But not all memories are helpful. Repeatedly recalling a traumatic event – such as an assault – can be harmful. About 1 in 3 people who experience severe trauma go on to develop post-traumatic stress disorder (PTSD), in which they re-live the traumatic event in the form of flashbacks and nightmares. Others develop panic disorder, phobias or depression.

Preventing this chain of events is challenging because fear memories form rapidly and last a long time. Current treatments involve re-exposing individuals to the traumatic event. This could be real-life exposure in the case of a phobia. Or it could involve visualizing the event, in the case of PTSD. Controlled re-exposure can help individuals learn new coping strategies. But it does not erase the initial fear memory.

A better approach might be to take advantage of the fact that new memories are unstable. To form a long-lasting memory trace, newly acquired information must go through a process called consolidation to stabilize it. This process takes place in an area of the brain called the hippocampus. If consolidation does not occur, new memory traces can fade away.

Li, Li et al. now show that preventing consolidation in the rat brain stops the animals from forming lasting memories of a stressful event, namely a foot shock. In the study, the rats first learned to associate a foot shock with a tone. This training took place inside a specific chamber. After learning the association, the rats began to freeze – a sign of fear – whenever they entered the chamber. This happened even if the tone was not played. But Li, Li et al. showed that they could reduce this fear response by activating cells in the hippocampus known as astrocytes, shortly after the learning episode.

Activating the astrocytes made them release a substance called adenosine. Molecules of adenosine then bound to and activated proteins called adenosine $A_1$ receptors. Administering a drug that activated these receptors directly had the same effect as activating the astrocytes themselves. This suggests that drugs of this type could one day help patients with fear-related disorders such as PTSD and phobias. For this to become a reality, new studies must test different drugs and find the best ways of administering them. After testing in animal models, the next step will be preliminary clinical trials in people.

*2016*). In humans, this can lead to various psychiatric disorders including post-traumatic stress disorder (PTSD), panic disorder, phobias, and depression (*Maren et al., 2013*; *Parsons and Ressler, 2013*). The estimated lifetime prevalence of fear- and stress-related disorders is close to 29% (*Kessler et al., 2005*). Yet limiting pathological fear is a considerable challenge since fear memories are rapidly acquired and temporally enduring. Fear extinction, such as exposure therapy, is a fundamental behavioral method to reduce fear and anxiety in humans. However, exposure therapy is a context-dependent learning process that does not erase the initial fear memory (*Parsons and Ressler, 2013*; *Tovote et al., 2015*). Fear memory could spontaneously recover or renew when patients are exposed to contexts similar or identical to those in which they first experienced trauma (*Britton et al., 2011*; *Maren, 2011*; *Tovote et al., 2015*). Therefore, it is urgent to find new therapeutic strategies for treating these disorders. Memory consolidation is a process by which newly acquired information is gradually stabilized by molecular and cellular processes after initial training (*Mednick et al., 2011*). New memories are labile and vulnerable to disruption during early consolidation (*Abraham and Williams, 2008*; *Dudai, 2004*; *Izquierdo et al., 1999*). Therefore, a feasible and more effective strategy for treating pathological fear would be to prevent fear memory consolidation soon after a traumatic event.

Context and cue processing are two major components of fear learning and memory. Moreover, context processing is essential for understanding the meaning of cues in a particular context. The dysregulation of contextual fear processing may lead to pathological fear-related disorders such as PTSD, phobias, panic disorder and depression (*Maren et al., 2013*; *Parsons and Ressler, 2013*). The hippocampus is thought to be critical in the formation of contextual fear memory (*Bast et al., 2003*; *Besnard et al., 2020*; *Daumas et al., 2005*; *Maren, 2001*; *Maren et al., 2013*). Hippocampal pyramidal neurons and interneurons have received much attention related to the contextual fear

memory process in the mammalian brain (*Lamsa and Lau, 2019*; *Roy et al., 2017*; *Xia et al., 2017*; *Zhu et al., 2014*). In vivo animal and human studies have found dynamic morphological and molecular changes in astrocytes during hippocampus-based contextual or spatial memory processes (*Choi et al., 2016*; *Sagi et al., 2012*), indicating the functional involvement of astrocytes in memory processes. However, as to whether and how astrocyte activity regulates contextual fear memory remains unclear.

It has been demonstrated that channelrhodopsin-2 (ChR2) expression is nontoxic, safe, stable, and functional (*Aravanis et al., 2007*; *Cardin et al., 2010*; *Deisseroth, 2011*; *Doroudchi et al., 2011*; *Zhang et al., 2006*). At present, ChR2 is widely used to explore the role of glia in regulating rodent behavior and circuits by precisely manipulating their $Ca^{2+}$ signaling (*Bang et al., 2016*; *Figueiredo et al., 2011*; *Gourine et al., 2010*; *Nam et al., 2016*; *Perea et al., 2014*; *Yamashita et al., 2014*). Our previous work showed that the $[Ca^{2+}]_i$ elevation induced by ChR2 in astrocytes is entirely from the extracellular space (*Yang et al., 2015*). Due to its proximity to the plasma membrane where exocytosis occurs, transmembrane $Ca^{2+}$ influx may be more efficient in inducing gliotransmitter release than $Ca^{2+}$ release from intracellular stores (*Chen et al., 2013*; *Tan et al., 2017*; *Yang et al., 2015*). Rats are well-adapted to the natural environment and are generally considered to perform well in learning and memory tasks (*Crawley, 2007*). The rat brain is relatively large, so injury from optical fibers is relatively small, thus photostimulation can be delivered specifically to the hippocampus. To clarify the exact role of astrocytes in fear memory and fear-related anxiety, we generated transgenic rats with astrocyte-specific ChR2 expression. We found that the optogenetic activation of astrocytes in CA1 within a critical time window after fear conditioning disrupted memory consolidation and persistently reduced contextual fear memory and fear-related anxiety. Conversely, reducing astrocyte $Ca^{2+}$ activity increased fear memory. Notably, our data revealed that the gliotransmitter adenosine and adenosine $A_1$ receptors ($A_1Rs$) were responsible for the fear memory attenuation and anxiolytic effect. Furthermore, intraperitoneal (i.p.) injection of the $A_1Rs$ agonist 2-chloro-N6-cyclopentyladenosine (CCPA) within the defined time window also decreased contextual fear memory and fear-related anxiety. Therefore, our findings demonstrate that astrocytes participate in the regulation of contextual fear memory through purinergic signaling. This provides a deeper understanding of the astrocyte-mediated regulation of fear memory and suggests an important therapeutic strategy against pathological fear-related disorders.

## Results

### Generation of transgenic rats with ChR2 uniquely expressed in astrocytes

ChR2 induces calcium elevation in astrocytes and it has been used as a tool for astrocyte activation (*Chen et al., 2013*; *Gourine et al., 2010*; *Tan et al., 2017*; *Yang et al., 2015*). To specifically manipulate the activity of astrocytes in the brain, we generated ChR2 knock-in rats (*GFAP-ChR2-EYFP*) with astrocyte-specific promoter-glial fibrillary acidic protein (GFAP). Co-staining of GFAP in EYFP-positive cells in the hippocampus verified the specific expression of ChR2 in astrocytes. We observed that 96.7% of the EYFP-positive astrocytes were GFAP-positive (*Figure 1A and B*). Furthermore, 92.5% of the GFAP-positive astrocytes in CA1 expressed EYFP (*Figure 1A and C*). We did not detect an EYFP signal in CA1 neurons (*Figure 1A*). We further confirmed these results with neuron and astrocyte co-culture. EYFP was only expressed in astrocytes but not neurons in vitro (*Figure 1—figure supplement 1A*). In addition, we also confirmed the specific expression of ChR2 in different brain areas such as the motor cortex, lateral posterior thalamic nucleus, and dorsomedial hypothalamic nucleus (*Figure 1—figure supplement 1B–D*).

To establish the functional response of astrocytes to optogenetic manipulation, $Ca^{2+}$ imaging and whole-cell recording were performed in rat hippocampal slices. After loading astrocytes with a $Ca^{2+}$ dye (Rhod-2 AM), we observed that blue light stimulation increased the $Ca^{2+}$ level in astrocytes expressing ChR2 (*Figure 1D and E*; *Figure 1—video 1*). Membrane depolarization was recorded in astrocytes that expressed ChR2-EYFP after light stimulation. We did not detect changes in the membrane potential in neurons with the same stimulation in these transgenic rats (*Figure 1F*). Astrocytes were further confirmed with different electrophysiological features; they showed a linear current-

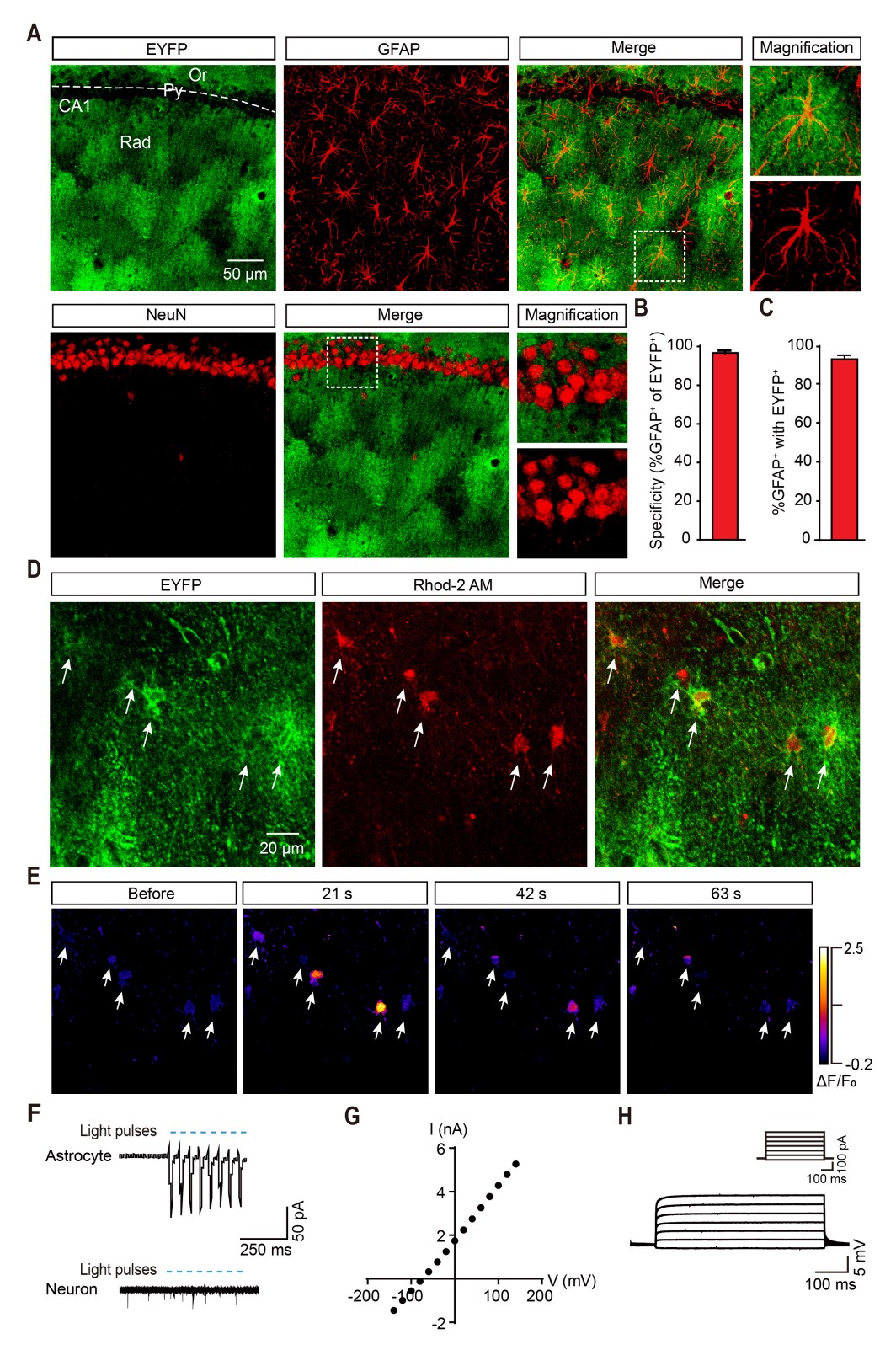

**Figure 1.** Specific ChR2 expression and light-induced $Ca^{2+}$ elevation in CA1 astrocytes of *GFAP-ChR2-EYFP* rats. (**A**) Immunohistochemistry (IHC) confocal images showing co-localization of EYFP labeling with the specific astrocytic marker GFAP, but not with the neuronal marker NeuN (right, higher magnification images; scale bar, 50 μm; Or, stratum oriens; Py, stratum pyramidale; Rad, stratum radiatum). (**B and C**) Quantification of the percentages of co-localization cells in astrocytes positive for EYFP (148/153 cells from six rats) and GFAP (148/160 cells from six rats). (**D**) Confocal

*Figure 1 continued on next page*

Figure 1 continued

images showing ChR2-expressing astrocytes loaded with the $Ca^{2+}$ fluorescent dye Rhod-2 AM. (E) Example time-lapse images of $Ca^{2+}$ signals before and 21, 42, and 63 s after the termination of light stimulation (488 nm, 10 s). (F) Light pulses (blue bars, 473 nm, 20 ms, 10 Hz) reliably induce inward currents in the ChR2-expressing astrocytes but not in neurons in the hippocampal slice. (G) Linear current-voltage relationship of an astrocyte in voltage-clamp mode. (H) Voltage responses from an astrocyte evoked by currents in steps of 120 pA from –100 pA to +680 pA. This figure has one figure and one video supplements.

The online version of this article includes the following video, source data, and figure supplement(s) for figure 1:

**Source data 1.** Data for *Figure 1*.
**Figure supplement 1.** ChR2-EYFP is specifically expressed in astrocytes.
**Figure 1—video 1.** Light stimulation increases $Ca^{2+}$ levels in astrocytes expressing ChR2.
https://elifesciences.org/articles/57155#fig1video1

voltage relationship under voltage-clamp mode, and no action potentials were detected even when they were depolarized to 0 mV (*Figure 1G and H*; *Figure 1—figure supplement 1E and F*).

## Astrocyte activation decreases contextual fear memory

To test the role of astrocytes in fear memory, we bilaterally implanted optical fibers into *GFAP-ChR2-EYFP* rats. To more comprehensively understand and to infer links with clinical symptoms (*McCarthy et al., 2017*), we examined the effects of astrocyte activation on fear memory in rats of both sexes. The fiber location in the hippocampus was confirmed after completing the experiments (*Figure 2A and B*). Photostimulation (473 nm, 10 Hz, 30 s on/30 s off, 15 min) in the stratum radiatum of dorsal CA1 followed the Pavlovian fear conditioning paradigm in which an initial tone stimulus (conditioned stimulus, CS, 30 s) co-terminated with a scrambled foot shock (unconditioned stimulus, US, 2 s) (*Figure 2C*; *Figure 2—figure supplement 1A*). Contextual and cued fear memory were measured on day 2 after fear acquisition (*Figure 2C*; *Figure 2—figure supplement 1A*). The photostimulated rats showed no differences in freezing levels during fear conditioning compared with controls (sham operation) (*Figure 2D*; *Figure 2—figure supplement 1B*). Contextual fear memory (*Figure 2E*; *Figure 2—figure supplement 1C*) but not cued fear memory (*Figure 2F*; *Figure 2—figure supplement 1D*) was significantly decreased in the photostimulated rats.

To further differentiate the role of astrocytes in memory consolidation and retrieval, rats were given photostimulation only during the recall test and showed freezing similar to controls (*Figure 2—figure supplement 2A–C*). These findings suggest that astrocyte photoactivation impairs memory consolidation, but not memory retrieval. To exclude the effect of light stimulation alone on freezing levels, littermate wild-type (WT) rats underwent the same protocol of photostimulation immediately after fear conditioning, and displayed freezing levels similar to sham-operated *GFAP-ChR2-EYFP* rats tested on day 2 (*Figure 2—figure supplement 2D–F*). We further tested the effects of longer photostimulation on fear memory attenuation. *GFAP-ChR2-EYFP* rats were fear conditioned as above (*Figure 2—figure supplement 3A*) and received repeated photostimulation (15 min, four times at 15-min intervals), which successfully attenuated their fear memory to levels comparable to those obtained after a single 15-min photostimulation (*Figure 2—figure supplement 3B and C*).

Astrocyte $Ca^{2+}$ signaling is attenuated by expression of the $Ca^{2+}$-extruder PMCA2w/b (plasma membrane $Ca^{2+}$ ATPase isoform two splice variant w/b) in hippocampal astrocytes (*Yu et al., 2020*; *Yu et al., 2018*). To investigate whether astrocyte activity is required for fear memory attenuation under normal conditions, we used an adeno-associated virus (AAV5) expressing human PMCA2w/b (hPMCA2w/b) with an astrocyte-specific GfaABC1D promoter to reduce hippocampal astrocyte $Ca^{2+}$ signaling (*Figure 2G–I*). In the control group, we microinjected tdTomato instead of hPMCA2w/b (*Figure 2—figure supplement 4A and B*). We ran behavioral tests 21 days after viral injection, and found that control and hPMCA2w/b rats did not differ in distance moved and center zone exploration time in the open-field test (OFT)(*Figure 2—figure supplement 4C and D*). To assess whether fear memory was affected by reducing hippocampal astrocyte $Ca^{2+}$ signaling, we used the Pavlovian fear conditioning paradigm again, and the contextual and cued fear memory were measured on day 2 after fear acquisition (*Figure 2C*). We found that control and hPMCA2w/b rats had comparable learning curves for fear conditioning (*Figure 2—figure supplement 4E*). Interestingly, contextual fear memory but not cued fear memory (*Figure 2J and K*) was significantly

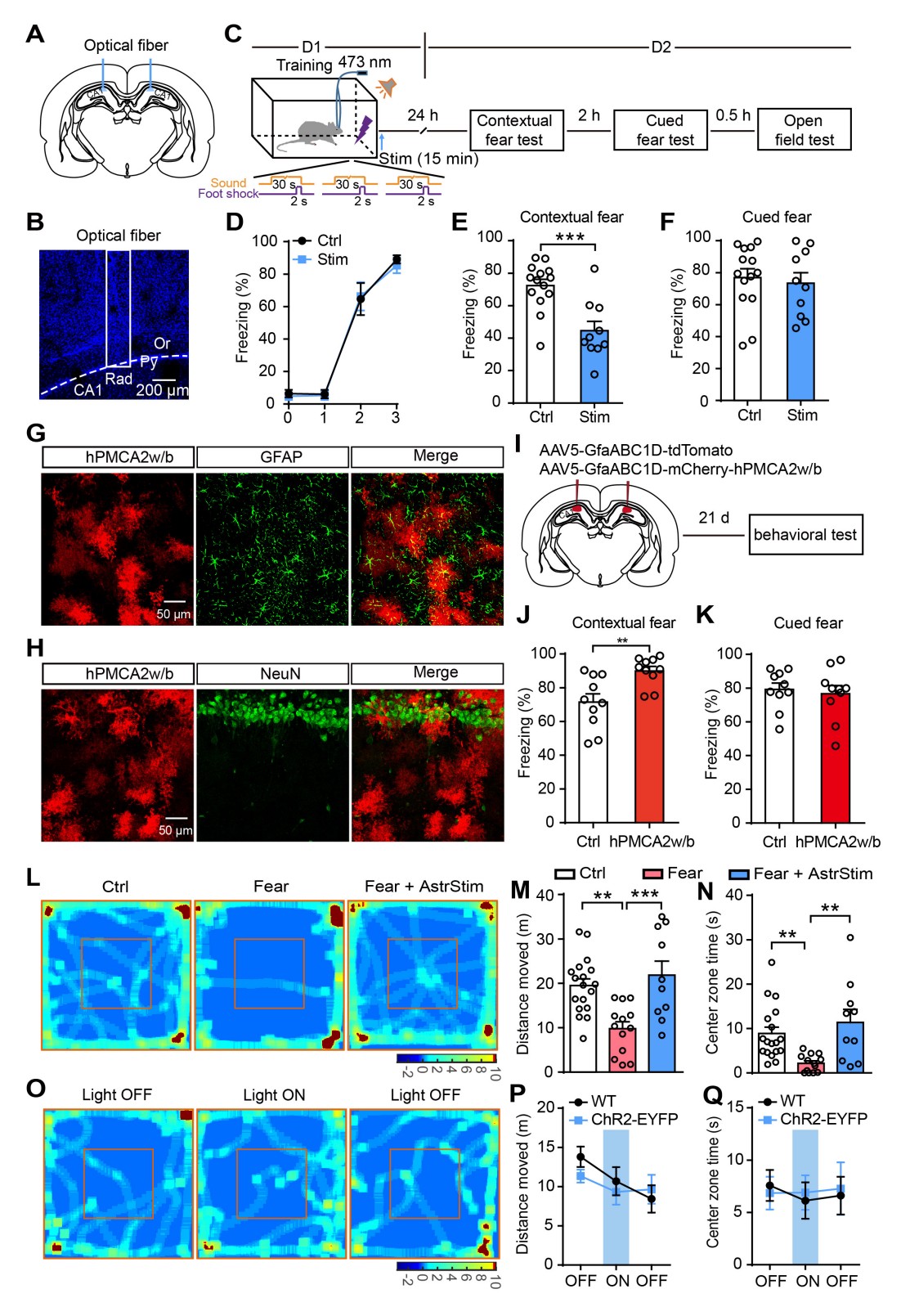

**Figure 2.** Optogenetic activation of astrocytes reduces fear memory and fear-related anxiety. (**A and B**) Schematic and confocal images of coronal sections showing the placement of optical fibers. (**C**) Schematic of the experimental design of fear conditioning, photostimulation, and the subsequent test protocols. (**D**) Freezing levels of control (sham operation, n = 14) and photostimulated *GFAP-ChR2-EYFP* rats (n = 10) during fear conditioning. (**E and F**) Freezing levels of control and photostimulated rats during contextual (E, p=0.0004, Student's unpaired two-tailed t-test) and cued fear tests (F, *Figure 2 continued on next page*

*Figure 2 continued*

p=0.4377) on day 2. (**G and H**) IHC confocal images showing co-localization of mCherry labeling with the specific astrocytic marker GFAP, but not with the neuronal marker NeuN. (**I**) AAV5 microinjection for expressing tdTomato and human PMCA2w/b (hPMCA2w/b) in hippocampal astrocytes. (**J and K**) Freezing levels of control (n = 10) and hPMCA2w/b (n = 10) rats during contextual fear memory (J, p=0.0044, Student's unpaired two-tailed t-test) and cued fear memory tests (K, p=0.6784). (**L**) Representative heatmaps of movement in the open field. Left, Ctrl, rats without fear conditioning; middle, Fear, rats with sham operation and fear conditioning; right, Fear+AstrStim, rats with photostimulation after fear conditioning. (**M**) Total distance moved of Ctrl (n = 18), Fear (n = 13), and Fear+AstrStim (n = 10) rats in the OFT (p=0.0018, p=0.0008, one-way ANOVA and Tukey's post-hoc test). (**N**) Center zone exploration time of Ctrl, Fear, and Fear+AstrStim rats in the OFT (p=0.0011, p=0.0033, Kruskal-Wallis and Dunn's post-hoc test). (**O**) Representative heatmaps of movement in the open field during light OFF–ON–OFF. (**P**) Total distance moved by WT and *GFAP-ChR2-EYFP* rats in the open field during light OFF-ON-OFF (group effect, $F_{(2, 78)}$=2.423, p=0.0953; epoch effect, $F_{(1, 78)}$=0.406, p=0.5259; interaction, $F_{(2, 78)}$=0.652, p=0.5238, two-way repeated-measures ANOVA and Bonferroni post-hoc test). (**Q**) Center zone exploration time of WT and *GFAP-ChR2-EYFP* rats in the open field during light OFF–ON–OFF (group effect, $F_{(2, 78)}$=0.05796, p=0.9437; epoch effect, $F_{(1, 78)}$=0.01926, p=0.89; interaction, $F_{(2, 78)}$=0.08292, p=0.9205, two-way repeated-measures ANOVA and Bonferroni post-hoc test). Error bars show the mean ± SEM. **p<0.01, ***p<0.001. This figure has four figure supplements.

The online version of this article includes the following source data and figure supplement(s) for figure 2:

**Source data 1.** Data for *Figure 2*.
**Figure supplement 1.** Optogenetic activation of astrocytes in female rats reduces fear memory.
**Figure supplement 1—source data 1.** Data for *Figure 2—figure supplement 1*.
**Figure supplement 2.** Photostimulation of *GFAP-ChR2-EYFP* rats during the recall test or photostimulation of WT rats after fear conditioning have no effect on contextual fear memory.
**Figure supplement 2—source data 1.** Data for *Figure 2—figure supplement 2*.
**Figure supplement 3.** Longer duration photostimulation induces a similar degree of fear memory attenuation.
**Figure supplement 3—source data 1.** Data for *Figure 2—figure supplement 3*.
**Figure supplement 4.** Specific tdTomato expression of control virus in hippocampal astrocytes and effects of hPMCA2w/b on behavioral tests.
**Figure supplement 4—source data 1.** Data for *Figure 2—figure supplement 4*.

increased in hPMCA2w/b rats. These results demonstrated that hippocampal astrocyte activity is indeed required for contextual fear memory.

## Astrocyte activation reduces anxiety-like behavior induced by fear conditioning

Severe traumatic stress leads to anxiety-like emotional responses, such as PTSD (*Britton et al., 2011*; *Parsons and Ressler, 2013*). To test the effect of astrocyte activation on fear-related anxiety-like behavior, we performed photostimulation immediately after fear conditioning. Thirty minutes after the cued fear test on day 2, we assessed the anxiety-like behavior of rats in the open field (*Figure 2C*).

The total moving distance (*Figure 2L and M*) and center zone exploration time (*Figure 2L and N*) of the fear-conditioned rats were significantly reduced compared with those of the controls (without fear conditioning), indicating that anxiety-like behavior was enhanced. Notably, photostimulation immediately after fear conditioning significantly rescued the decreased moving distance (*Figure 2L and M*) and center zone exploration time (*Figure 2L and N*), indicating that astrocytic activation exerts an anxiolytic effect.

We then assessed whether the motor performance was directly affected during photostimulation using the OFT. We used 3-min epochs as the paradigm for light OFF–ON–OFF. Photostimulation of astrocytes in CA1 did not affect the distance moved and center zone exploration time (*Figure 2O–Q*).

## A critical time-window exists between fear conditioning and astrocyte activation for fear memory attenuation

To determine whether there is a time-window of astrocyte activation after training for the disruption of fear memory consolidation, rats were photostimulated for 15 min in CA1 1–3 hr after fear conditioning (*Figure 3A*), and the contextual fear memory was assessed 24 hr later. Control and photostimulated rats had comparable learning curves for fear conditioning (*Figure 3B*). Photostimulation of astrocytes 1 hr, but not 2 hr or 3 hr (*Figure 3C*) after fear conditioning resulted in a significantly decreased fear response, indicating that astrocyte activation within a critical time window after

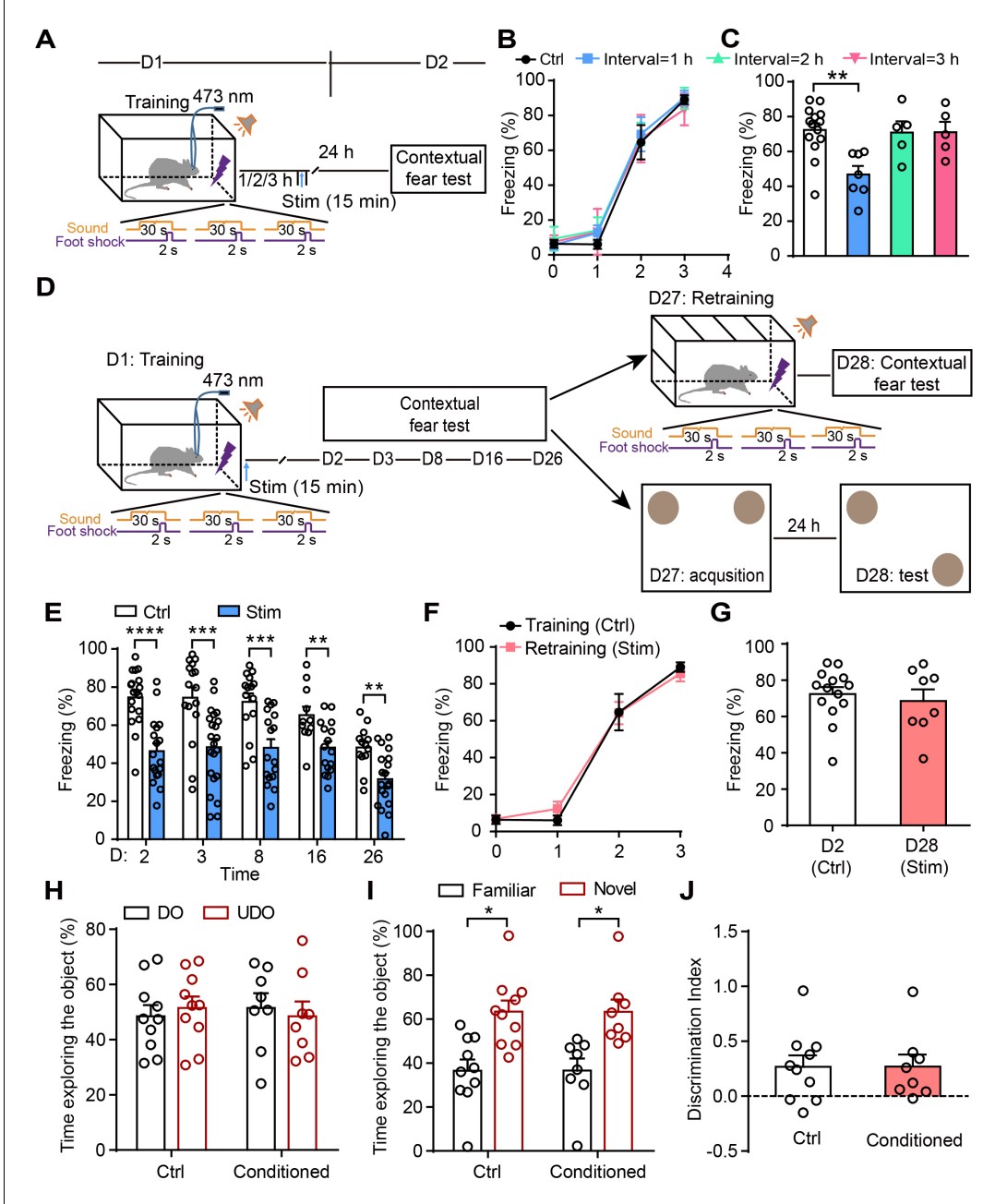

**Figure 3.** Astrocyte activation disrupts consolidation within a critical time-window and induces long-lasting contextual fear memory attenuation. (A and D) Schematic of the experimental design of fear conditioning, photostimulation, and the subsequent test protocols. (B) Freezing levels of control (sham operation without photostimulation, n = 14) and photostimulated (Interval = 1 hr, n = 7; Interval = 2 hr, n = 5; Interval = 3 hr, n = 5) *GFAP-ChR2-EYFP* rats during fear conditioning. (C) Freezing levels of control and photostimulated *GFAP-ChR2-EYFP* rats during contextual fear tests on day 2 (p=0.0027, one-way ANOVA Tukey's post hoc test). (E) Freezing levels of control and photostimulated *GFAP-ChR2-EYFP* rats during contextual fear tests on day 2 (n = 18 ctrl, n = 18 stim; p < 0.0001, Student's unpaired two-tailed t-test), day 3 (n = 16 ctrl, n = 22 stim; p=0.0006), day 8 (n = 15 ctrl, n = 17 stim; p=0.0006), day 16 (n = 11 ctrl, n = 16 stim; p=0.0051), and day 26 (n = 13 ctrl, n = 19 stim, p=0.0017). (F) Freezing levels of control rats conditioned on day 1 and retrained rats conditioned on day 27 (n = 8). (G) Freezing levels of control rats tested on day 2 and retrained rats tested on day 28 (p=0.5929, Student's unpaired two-tailed t-test). (H) Time spent exploring the displaced object (DO) and undisplaced object (UDO) by control (n = 10) and conditioned rats (n = 8) during the sample phase (ctrl, p=0.7170; conditioned, p=0.7871, Student's paired two-tailed t-test). (I) Time spent exploring the objects in familiar and novel locations by control and conditioned rats during the test phase (ctrl, p=0.0271; conditioned, p=0.0468, Student's paired two-tailed t-test). (J) The discrimination index of control and conditioned rats measured during the test phase (p=0.9961, Student's unpaired two-tailed t-test). Error bars show the mean ± SEM. *p<0.05, **p<0.01, ***p<0.001, ****p<0.0001.

The online version of this article includes the following source data for figure 3:

**Source data 1.** Data for *Figure 3*.

traumatic events is more efficient for disrupting memory consolidation and reducing contextual fear memory.

## Astrocyte activation induces long-lasting contextual fear memory attenuation

To measure whether the contextual fear memory attenuation induced by astrocyte activation is long-lasting, we assessed it on days 2, 3, 8, 16, and 26 after fear acquisition (*Figure 3D*). Compared with the controls (sham operation), the rats receiving photostimulation immediately after fear acquisition showed a persistent decrease of contextual fear memory lasting up to 26 days (*Figure 3E*). On day 27, some of the photostimulated rats were re-trained in a new conditioning chamber and exhibited a learning curve comparable to controls conditioned on day 1 (*Figure 3F*). The other photostimulated rats (conditioned) performed the object location recognition (OLR) task and the percentages of time spent exploring the objects were similar to controls in the sample phase (*Figure 3H*), indicating that astrocyte activation does not affect new learning. Furthermore, retrained rats had normal memory retention on day 28 during contextual fear test compared with controls on day 2 (*Figure 3G*). In the test phase of the OLR on day 28, the conditioned rats spent more time exploring the object in a novel location relative to the object in a familiar location (*Figure 3I*) and the discrimination index was similar to controls (*Figure 3J*). All these data suggested that astrocyte activation does not affect new memory formation. These results reveal that the astrocyte activation reliably and persistently attenuates temporally coupled fear memory.

## Astrocyte activation attenuates fear memory through A$_1$Rs

Previous studies in our lab (*Chen et al., 2013*; *Tan et al., 2017*) and those of other investigators *Pascual et al., 2005* found that the ATP derived from astrocytes degrades to adenosine and leads to the suppression of nearby synapses. To determine whether the photostimulation of astrocytes triggered increased ATP and adenosine concentrations in CA1 of *GFAP-ChR2-EYFP* rats, we performed in vivo microdialysis experiments (*Figure 4A*).

Following the collection of samples of baseline dialysate, we initiated the second phase of microdialysis starting with 15 min of photostimulation. The ATP and adenosine concentrations in the interstitial fluid were significantly higher (ATP: 1st, $0.74 \pm 0.26$ nM, 2nd, $1.98 \pm 0.38$ nM; adenosine: 1st, $2.88 \pm 0.17$ μM, 2nd, $3.42 \pm 0.11$ μM) in the photostimulation phase (*Figure 4B*), demonstrating that astrocyte activation induces ATP release in the hippocampus. To determine whether the fear memory attenuation induced by astrocyte activation was mediated by ATP or its degradation product, adenosine, we implanted infusion cannulae bilaterally into dorsal CA1 (*Figure 4C*). ATP-γ-S, the specific P2Y receptor agonist 2-(methylthio)adenosine 5'-diphosphate (MesADP), or vehicle were separately injected into the dorsal CA1 bilaterally after fear conditioning, then we assessed contextual fear memory on day 2 (*Figure 4D*). The vehicle and drug-treated rats had comparable learning curves for fear conditioning (*Figure 4E–G*). There was no difference in contextual fear memory (*Figure 4H*). Further, agonists of adenosine receptors were bilaterally delivered to the dorsal CA1 immediately after fear conditioning. The rats given the non-hydrolyzable adenosine analog 5'-(N-ethylcarboxamido)adenosine (NECA) and the specific A$_1$R agonist CCPA displayed significantly decreased contextual fear memory on day 2 compared with vehicle-treated rats (*Figure 4I*), while the specific adenosine A$_{2A}$ receptor (A$_{2A}$R) agonist CGS 21680 hydrochloride had no effect (*Figure 4J*). NECA and CCPA in the dorsal CA1 did not affect spontaneous locomotor activity or center zone exploration time in the OFT (*Figure 4—figure supplement 1A–E*). These results showed that the activation of A$_1$Rs after memory acquisition mimicked the effects of astrocyte activation on contextual fear memory, indicating that A$_1$Rs mediate the contextual fear memory attenuation.

To further confirm that adenosine participates in the astrocyte activation-induced attenuation of contextual fear memory, we separately injected into CA1 the ectonucleotidase inhibitor ARL 67156 trisodium salt hydrate, which prevents ATP from converting into adenosine, the specific A$_1$R antagonist 8-cyclopentyl-1,3-dimethylxanthine (CPT), or the specific A$_{2A}$R antagonist SCH 58261. Vehicle and drug treatment were paired with photostimulation of astrocytes in CA1 immediately after fear conditioning, and contextual fear memory was examined on day 2 (*Figure 5A and B*). Control rats (without photostimulation or pharmacological treatment), photostimulated rats (without pharmacological treatment), and photostimulated rats paired with vehicle or drug treatment had comparable

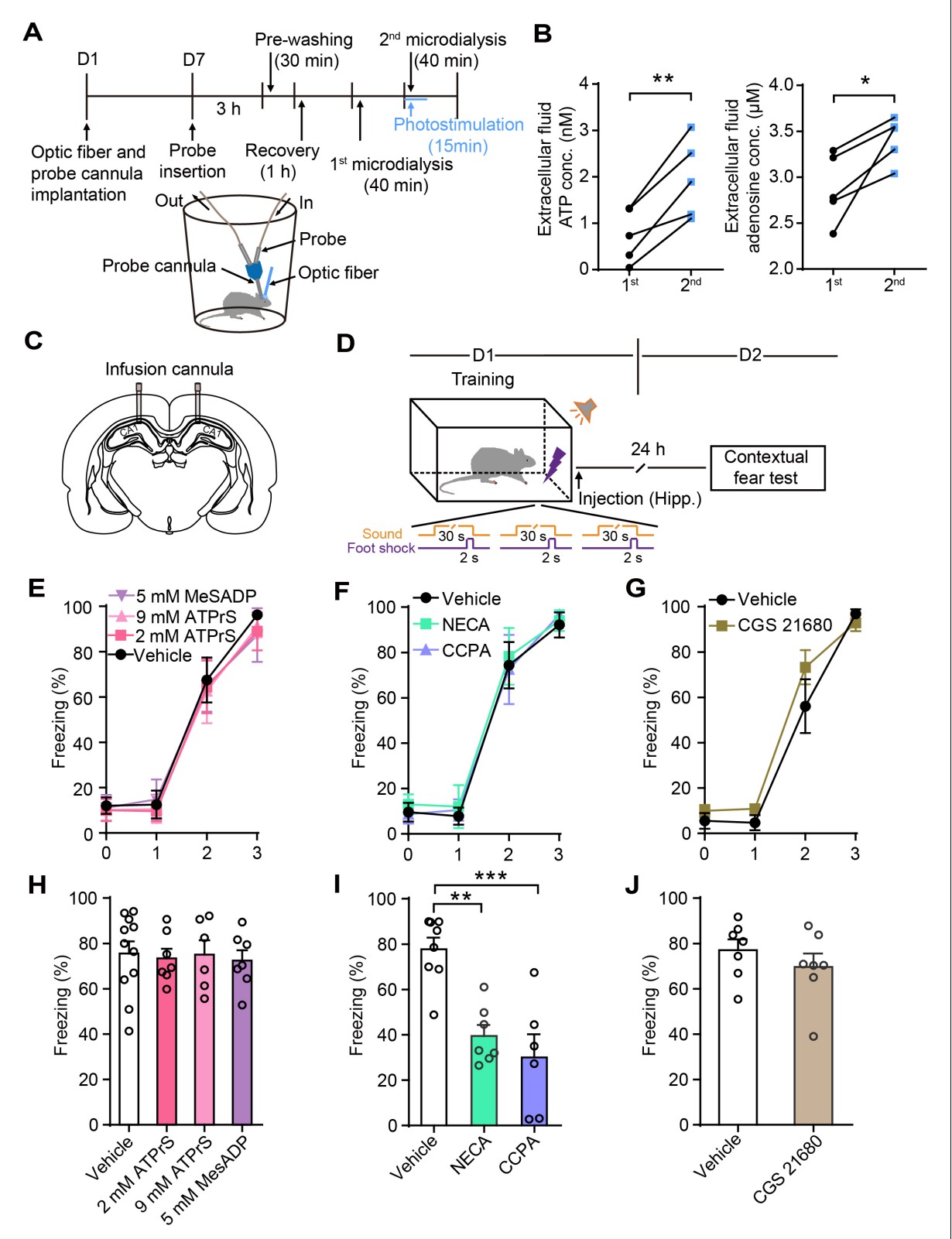

**Figure 4.** Astrocyte activation in CA1 induces ATP release and $A_1$Rs mediate contextual fear memory attenuation. (**A**) Schematic of the experimental design of microdialysis in vivo. (**B**) Extracellular ATP and adenosine concentrations in the dialysate prior to and following photostimulation (n = 5; p=0.0058, p=0.0028, Student's paired two-tailed t-test). (**C**) Schematic showing the placement of implanted cannulae. (**D**) Schematic of the experimental design of fear conditioning, drug administration, and the subsequent test protocols. (**E–G**) Freezing levels of vehicle and drug-treated rats during fear

*Figure 4 continued on next page*

Figure 4 continued

conditioning (E, n = 11 Vehicle, n = 7 ATP-γ-S (2 mM), n = 6 ATP-γ-S (9 mM), n = 7 MesADP (5 mM); F, n = 8 Vehicle, n = 7 NECA, n = 6 CCPA; G, n = 7 Vehicle, n = 7 CGS 21680). (H) Freezing levels of vehicle, ATP-γ-S, and MesADP-treated rats during contextual fear tests on day 2 (p>0.05, one-way ANOVA Tukey's post-hoc test). (I) Freezing levels of rats treated with vehicle, NECA (2 mM), and CCPA (5 mM) during contextual fear tests on day 2 (p=0.0015, p=0.0002, one-way ANOVA and Tukey's post hoc test). (J) Freezing levels of vehicle and CGS 21680 (5 mM)-treated rats during contextual fear tests on day 2 (p=0.3481, Student's unpaired two-tailed t-test). Error bars show the mean ± SEM. *p<0.05, **p<0.01, ***p<0.001. This figure has one figure supplement.

The online version of this article includes the following source data and figure supplement(s) for figure 4:

**Source data 1.** Data for *Figure 4*.
**Figure supplement 1.** NECA and CCPA in the hippocampus do not affect spontaneous locomotor activity and center zone exploration time of the OFT.
**Figure supplement 1—source data 1.** Data for *Figure 4—figure supplement 1*.

learning curves for fear conditioning (*Figure 5C–E*). Compared to controls, astrocyte activation alone or paired with vehicle significantly decreased contextual fear memory to a comparable level (*Figure 5F–H*). Astrocyte activation paired with ARL 67156 reversed the freezing level to that of controls, showing that the attenuation effect of astrocyte photostimulation on fear memory was almost totally blocked by ARL 67156 (*Figure 5F*). These results indicate that the fear memory attenuation induced by astrocyte activation is abrogated with diminished levels of adenosine. CPT treatment also dramatically blocked the attenuation of fear memory induced by astrocyte photostimulation (*Figure 5G*), indicating that astrocyte activation attenuates fear memory in an $A_1R$-dependent manner. Notably, CPT in CA1 did not affect the spontaneous locomotor activity and center zone exploration time in the OFT (*Figure 5—figure supplement 1A–C*). The specific $A_{2A}R$ antagonist SCH 58261 failed to block the attenuation of fear memory induced by astrocyte activation (*Figure 5H*). Altogether, these results suggest that $A_1Rs$ are necessary for the contextual fear memory attenuation induced by astrocyte activation.

## Intraperitoneal injection of CCPA within a defined time window decreases contextual fear memory and fear-related anxiety

To further determine the therapeutic applicability of CCPA for fear memory attenuation, we tested whether the i.p. injection of CCPA decreases fear memory. Rats were injected with vehicle or CCPA immediately after fear conditioning and the contextual fear memory was assessed on days 2 and 3 (*Figure 6A*). Rats with vehicle and CCPA injection had comparable learning curves for fear conditioning (*Figure 6—figure supplement 1A–C*). Rats injected with vehicle or CCPA (0.03 mg/kg) did not differ in contextual fear memory tested on days 2 and 3 (*Figure 6B*). However, rats injected with higher doses of CCPA (0.1 and 0.3 mg/kg) showed significantly attenuated contextual fear memory tested on days 2 and 3, compared to rats with vehicle injection (*Figure 6C and D*). To determine whether there is also a time-window of CCPA injection after training for the disruption of fear memory consolidation, rats were injected with vehicle or CCPA (0.1 mg/kg) 0.5–2 hr after fear conditioning, and the contextual fear memory was assessed on days 2 and 3 (*Figure 6E*). Rats with vehicle and CCPA injection had comparable learning curves for fear conditioning (*Figure 6—figure supplement 1D–F*). CCPA injection 0.5 hr and 1 hr (*Figure 6F and G*), but not 2 hr (*Figure 6H*) after fear conditioning significantly reduced the fear response. These results show that CCPA injection within a critical time-window after a traumatic experience disrupts memory consolidation and reduces contextual fear memory.

To test whether CCPA affects fear-related anxiety-like behavior, rats were injected with vehicle or CCPA (0.1 mg/kg, i.p.) immediately after fear conditioning, and tested in the open field 3 hr or 24 hr later. Compared with controls (without fear conditioning), the fear-conditioned rats with vehicle injection showed a decreased moving distance and center zone exploration time in the OFT, while CCPA injection significantly rescued the decreased moving distance and center zone exploration time (*Figure 6I–N*). Notably, CCPA injection at 0.1 mg/kg did not affect the motor performance and center zone exploration time of unconditioned rats in the OFT (*Figure 6—figure supplement 2A–C*). CCPA injection at the higher concentrations of 0.3 mg/kg decreased the moving distance in the OFT (*Figure 6—figure supplement 2A and B*). However, the motor performance recovered 5 hr after higher doses CCPA injection (*Figure 6—figure supplement 2D*).

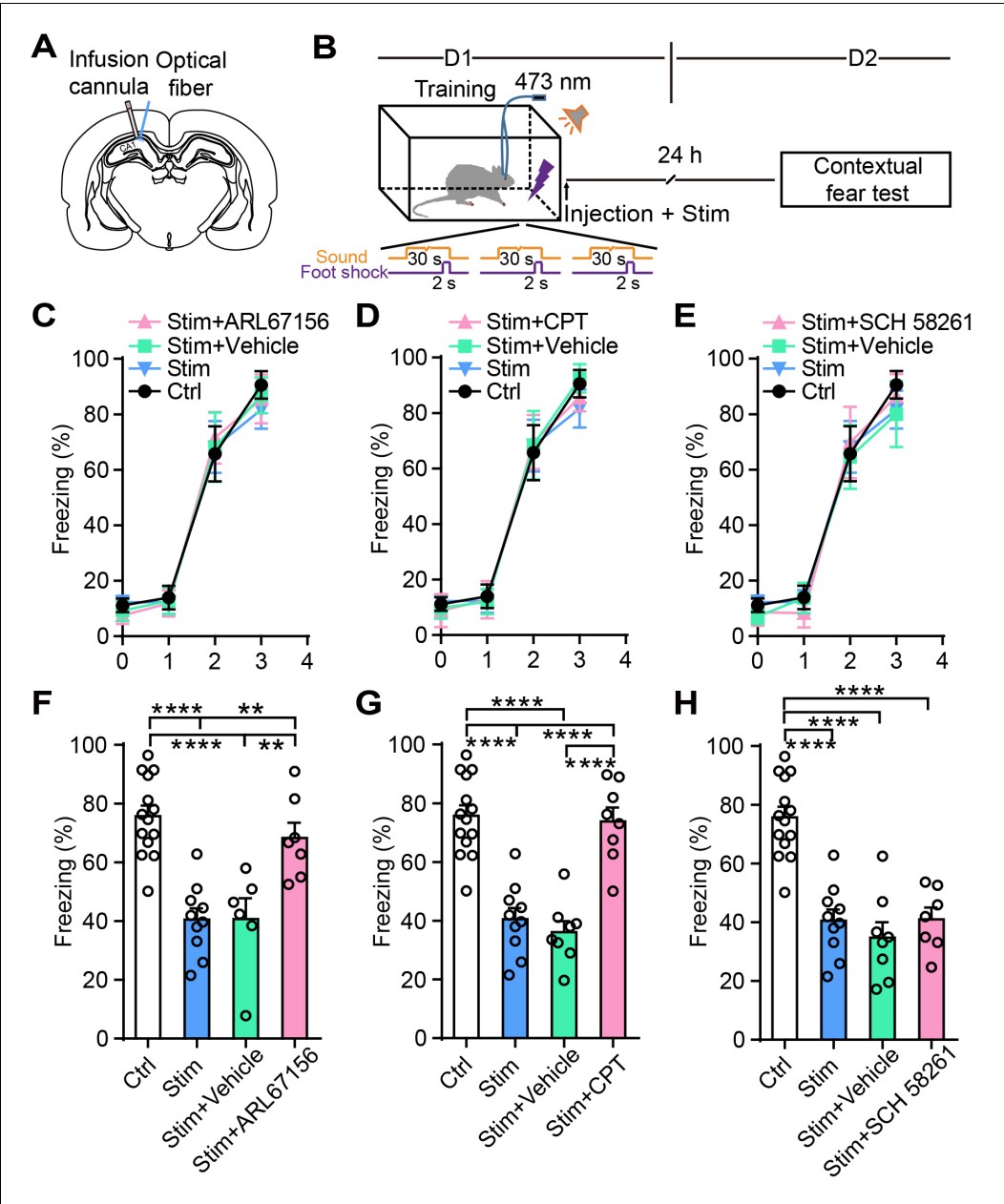

**Figure 5.** Ectonucleotidase inhibitor and $A_1R$ antagonist block the attenuation of the fear memory induced by astrocyte activation. (**A**) Schematic showing the placement of the optical fiber and cannula implant. (**B**) Schematic of the experimental design of fear conditioning, photostimulation, drug administration, and the subsequent test protocols. (**C–E**) Freezing levels of control (no photostimulation or pharmacological treatment), Stim (photostimulation), Stim+Vehicle (photostimulation paired with vehicle treatment), and Stim+drug (photostimulation paired with ARL67156, CPT, or SCH 58261) rats during fear conditioning. (**F**) Freezing levels of control (n = 14), Stim (n = 10), Stim+Vehicle (n = 6), and Stim+ARL67156 (30 mM, n = 7) rats during contextual fear tests on day 2 (p<0.0001, p=0.0015, p=0.0055, one-way ANOVA and Tukey's post-hoc test). (**G**) Freezing levels of control, Stim, Stim+Vehicle (n = 8), and Stim+CPT (n = 8, 1 mM) rats during contextual fear tests on day 2 (p<0.0001, one-way ANOVA and Tukey's post hoc test). (**H**) Freezing levels of control, Stim, Stim+Vehicle (n = 8), and Stim+SCH 58261 (n = 7, 1 mM) rats during contextual fear tests on day 2 (p<0.0001, one-way ANOVA and Tukey's post-hoc test). Error bars show the mean ± SEM. **p<0.01, ****p<0.0001. This figure has one figure supplement.

The online version of this article includes the following source data and figure supplement(s) for figure 5:

**Source data 1.** Data for *Figure 5*.

*Figure 5 continued on next page*

*Figure 5 continued*

**Figure supplement 1.** CPT in the hippocampus do not affect spontaneous locomotor activity and center zone exploration time of the OFT.

**Figure supplement 1—source data 1.** Data for *Figure 5—figure supplement 1*.

## Discussion

Accumulating evidence suggest that astrocyte activity is crucial for synaptic regulation and plasticity, which are considered to be involved in learning and memory (*Chen et al., 2013*; *Martin-Fernandez et al., 2017*; *Tan et al., 2017*; *Yang et al., 2003*; *Zhang et al., 2003*). Previous studies have reported the essential role of astrocytes in memory processing (*Gao et al., 2016*; *Robin et al., 2018*; *Suzuki et al., 2011*; *Vignoli et al., 2016*). However, it remains unclear whether and how astrocyte activity regulates contextual fear memory. We provide evidence for the first time that astrocyte activation by precise optogenetic stimulation within a defined time window (1 hr) disrupts memory consolidation and dramatically reduces contextual fear memory and the related anxiety. The fear memory attenuation is robust and persistent, while learning capacity remains intact. On the contrary, reducing $Ca^{2+}$ activity in astrocytes increased fear memory. Our further results reveal that astrocyte activation increases the extracellular ATP and adenosine concentrations, and the fear memory attenuation and anxiolytic effect are mediated by $A_1R$ activation. Thus, our findings provide a deeper understanding of astrocyte-mediated regulation of fear memory and the complexity of the functional consequences of astrocyte regulation.

The hippocampus is essential for the formation of contextual fear memory (*Bast et al., 2003*; *Daumas et al., 2005*; *Maren, 2001*; *Maren et al., 2013*), and hippocampal lesions selectively affect contextual fear memory but not cued fear memory (*Kim and Fanselow, 1992*; *Phillips and LeDoux, 1992*). Context plays a central role in understanding the meaning of cues, abstracting the surrounding information and anticipating the future in a particular context. Disorders in contextual processing may lead to fear-related diseases, including PTSD, panic disorder, phobias and depression (*Maren et al., 2013*; *Parsons and Ressler, 2013*). PTSD may be the most representative of context-processing diseases, given that its core features involve recurrent and intrusive recollections (flashbacks) of an experienced traumatic event (*Rosenbaum, 2004*). Finding new mechanisms underlying contextual memory processing is thus critical in developing strategies for relieving sufferers from pathological fear. Memory consolidation is a molecular and cellular processes by which newly-acquired information is gradually stabilized by strengthening synaptic connections after the initial training (*Dudai, 2004*). It has been proposed that long-term potentiation (LTP), a form of synaptic plasticity, is an intrinsic property of this consolidation (*Abraham and Williams, 2008*; *Kandel et al., 2014*; *Yang et al., 2018*). Once LTP or memory has passed into the protein synthesis-dependent phase, LTP is highly resistant to disruption, while memory becomes more difficult to erase and is said to have been consolidated (*Abraham and Williams, 2008*; *Lu et al., 2018*; *Mednick et al., 2011*). So if the time between learning and interference is sufficiently long for the processes associated with LTP to have occurred, the memory trace would be less vulnerable. Our present study showed that photoactivation of astrocytes in CA1 within the early memory consolidation phase (1 hr) but not beyond it (2 hr, 3 hr or during retrieval) after a fear stimulus produced a long-lasting attenuation of contextual fear memory and an anxiolytic effect. This finding is in accordance with reports that new memories are sensitive to interference within a short time (1 hr) after learning, but not after a relatively long interval (6 hr) during which the memory trace becomes consolidated (*Mednick et al., 2011*). More importantly, our results suggest a new strategy of astrocyte-based disruption of memory consolidation, which leads to persistent fear memory attenuation with attenuated spontaneous recovery of fear memory.

Astrocytes respond to neuronal activity and release different neuroactive molecules, among which ATP (adenosine), glutamate, D-serine, and gamma-aminobutyric acid are the major gliotransmitters identified as regulators of synaptic transmission (*Araque et al., 2014*). Recent study showed that activating astrocytes by chemogenetic or optogenetic recruitment of their Gq-coupled signaling before or during memory acquisition induces N-methyl-D-aspartate-dependent **long-term potentiation** in CA1 through the release of D-serine and enhances memory acquisition (*Adamsky et al., 2018*). However, another study showed that chemogenetic activation of astrocytes before memory

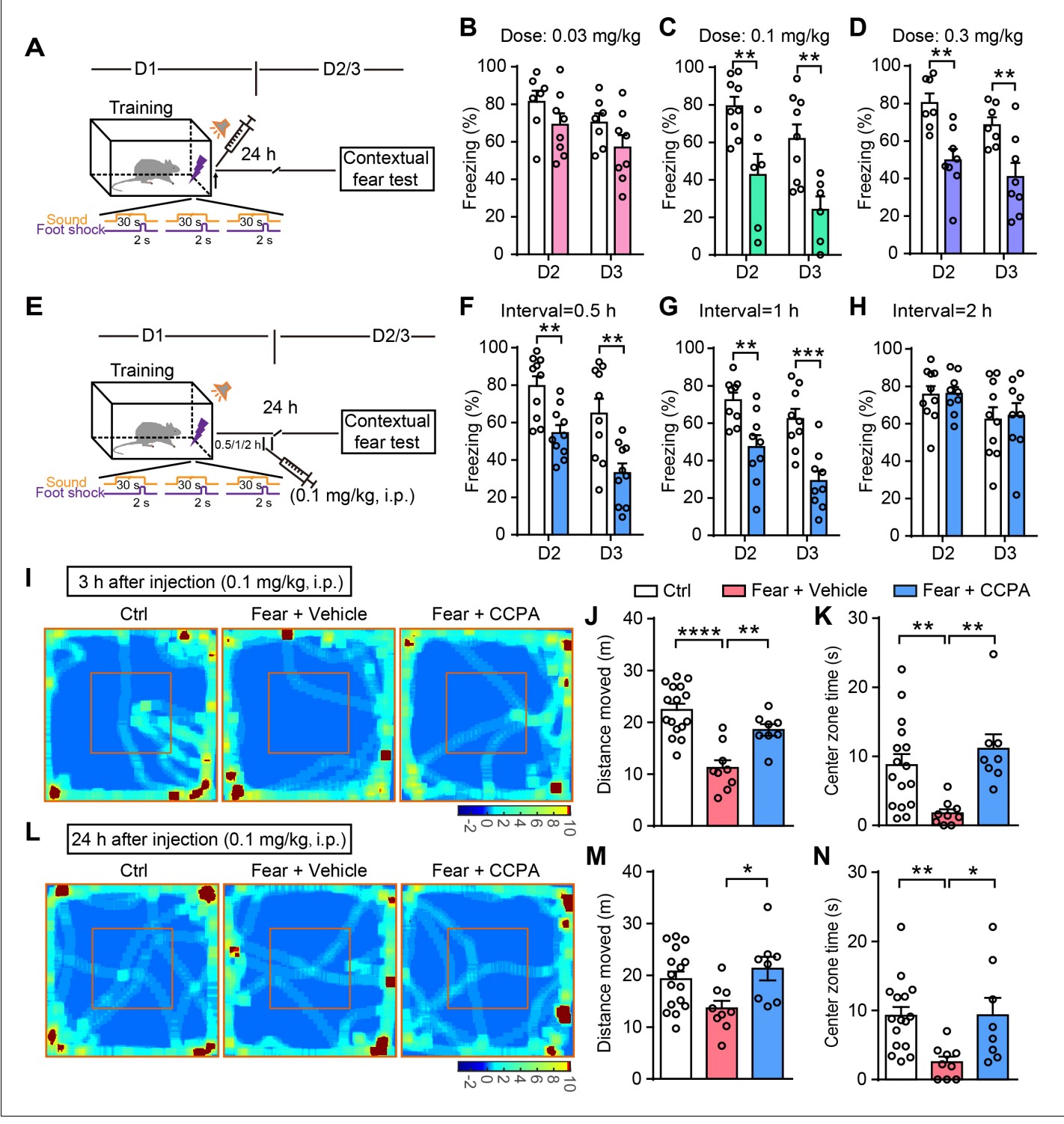

**Figure 6.** Administration of CCPA (i.p.) within a defined time window reduces contextual fear memory and fear-related anxiety. (**A and E**) Schematic of the experimental design of fear conditioning, drug administration, and the subsequent test protocols. (**B–D**) Freezing levels of rats with different doses of vehicle and CCPA treatment during contextual fear tests on day 2 (B, n = 7 Vehicle, n = 8 CCPA, p=0.1765; C, n = 9 Vehicle, n = 6 CCPA, p=0.0053; D, n = 7 Vehicle, n = 8 CCPA, p=0.0021, Student's unpaired two-tailed t-test) and day 3 (B, p=0.1409; C, p=0.0048; D, p=0.0086). (**F–H**) Freezing levels of rats with vehicle and CCPA treatment at different time tested on day 2 (F, n = 10 Vehicle, n = 10 CCPA, p=0.0016; G, n = 9 Vehicle, n = 9 CCPA, p=0.0041; H, n = 10 Vehicle, n = 9 CCPA, p=0.9417, Student's unpaired two-tailed t-test) and day 3 (F, p=0.0033; G, p=0.0004; H, p=0.8366). (**I and L**) Representative heatmaps of movement in the open field 3 hr and 24 hr after CCPA injection (0.1 mg/kg). Left, Ctrl, rats without fear conditioning; middle, Fear+Vehicle, fear-conditioned rats with vehicle injection; right, Fear+CCPA, fear-conditioned rats with CCPA injection. (**J and M**) Total distance

*Figure 6 continued on next page*

*Figure 6 continued*

moved by Control (n = 16), Fear+Vehicle (n = 9) and Fear+CCPA (n = 8) rats in the OFT 3 hr and 24 hr after injection (J, 3 hr, p<0.0001, p=0.0031, one-way ANOVA and Tukey's post hoc test; M, 24 hr, p=0.022). (K and N) Center zone exploration time of Control, Fear+Vehicle, and Fear+CCPA rats in the OFT 3 hr and 24 hr after injection (K, 3 hr, p=0.0069, p=0.0014, Kruskal-Wallis and Dunn's post hoc test; N, 24 hr, p=0.0025, p=0.041). Error bars show the mean ± SEM. *p<0.05, **p<0.01, ***p<0.001, ****p<0.0001. This figure has two figure supplements.

The online version of this article includes the following source data and figure supplement(s) for figure 6:

**Source data 1.** Data for *Figure 6*.
**Figure supplement 1.** Rats with vehicle and CCPA injection have comparable learning curves for fear conditioning.
**Figure supplement 1—source data 1.** Data for *Figure 6—figure supplement 1*.
**Figure supplement 2.** Intraperitoneal injection of CCPA at a lower effective dose (0.1 mg/kg) does not affect locomotor and center zone exploration time in the OFT.
**Figure supplement 2—source data 1.** Data for *Figure 6—figure supplement 2*.

retrieval reduces the expression of an acquired cued fear response, which is mediated by inhibiting excitatory synapses from the basolateral amygdala via $A_1R$ activation and enhancing inhibitory synapses from the lateral subdivision of the central amygdala via **$A_{2A}$** activation (*Martin-Fernandez et al., 2017*). These contradictory results indicate that different patterns of stimulation may induce astrocytes to release different substances which then activate their associated receptors on nearby neurons, leading to different forms of synaptic regulation (*Covelo and Araque, 2018*).

Previous work in our lab established that the gliotransmitters ATP and adenosine but not glutamate and D-serine are involved in astrocyte-mediated synaptic suppression (*Chen et al., 2013*; *Tan et al., 2017*; *Zhang et al., 2003*). Astrocyte-derived ATP increases the activity of cholecystokinin-expressing interneurons through the activation of P2Y1 receptors and decreases pyramidal neuron activity through the activation of $A_1Rs$ in CA1, resulting in the downregulation of the whole network activity in the hippocampus (*Tan et al., 2017*). Similarly, a recent publication showed that adenosine degraded from the astrocyte-derived ATP upregulates the synaptic inhibition of pyramidal neurons by somatostatin-expressing interneurons via $A_1R$ activation (*Matos et al., 2018*). The activation of $A_1Rs$ inhibits adenylate cyclase, which causes a decrease in the second messenger cAMP, inhibits voltage-gated $Ca^{2+}$ channels and activates G-protein-coupled inwardly rectifying $K^+$ channels through Gi/o βγ-subunits, and decreases the excitability of pyramidal neurons (*Burnstock et al., 2011*; *Fields and Burnstock, 2006*; *Tan et al., 2017*; *Wetherington and Lambert, 2002*). Previous studies demonstrate that neurons with relatively high intrinsic excitability promote memory integration (via co-allocation to overlapping engrams), while decreased excitability promotes memory separation (*Josselyn, 2010*; *Josselyn and Frankland, 2018*). Our in vivo microdialysis experiments showed that photostimulation of astrocytes led to significant increase in extracellular ATP and adenosine concentrations. Both an ectonucleotidase inhibitor (which prevents ATP from converting to adenosine) and a specific $A_1R$ antagonist dramatically reversed astrocyte activation-induced fear memory attenuation. But a specific $A_{2A}R$ antagonist did not have this effect. Furthermore, $A_1R$ agonist application within the critical time window mimicked the effects of astrocyte activation on fear memory and the anxiolytic effect. Therefore, we demonstrate that the astrocyte-mediated attenuation of contextual fear memory and the related anxiety found in our experimental conditions depends on $A_1Rs$, which could act as potential new molecular targets for the treatment of fear-related disorders.

Together, our results identify a functional role of astrocytes in contextual fear memory and the related anxiety and reveal that astrocytes are essential elements in fear memory processing through purinergic signaling in the hippocampus. Consequently, our findings provide a deeper understanding of astrocyte-mediated regulation of fear memory and suggest a new and important therapeutic strategy against pathological fear-related disorders. An attempt to use this strategy in clinic could be very promising.

# Materials and methods

**Key resources table**

*Continued on next page*

*Continued*

| Reagent type (species) or resource | Designation | Source or reference | Identifiers | Additional information |
|---|---|---|---|---|
| Reagent type (species) or resource | Designation | Source or reference | Identifiers | Additional information |
| Strain, strain background (*Rattus norvegicus*, male/female) | Sprague-Dawley rats | Shanghai SLAC Laboratory Animal C.,Ltd | N/A | |
| Strain, strain background (*R. norvegicus*, male/female) | *GFAP-ChR2-EYFP* rats | Institute of Neuroscience, Chinese Academy of Sciences | N/A | |
| Genetic reagent (virus) | pZac2.1 GfaABC1D mCherry-hPMCA2w/b | Addgene | Cat# 111568 | |
| Genetic reagent (virus) | AAV5 GfaABC1D mCherry-hPMCA2w/b | Vigene Biosciences | N/A | |
| Genetic reagent (virus) | AAV5 GfaABC1D tdTomato | Vigene Biosciences | N/A | |
| Antibody | Anti-GFAP, rabbit polyclonal | Millipore | Cat# AB5804 | (1:500) |
| Antibody | Anti-NeuN, mouse monoclonal | Millipore | Cat# MAB377 | (1:400) |
| Antibody | Anti-DsRed, rabbit polyclonal | Clontech | Cat# 632496 | (1:500) |
| Antibody | Alexa Fluor 568 donkey anti-rabbit IgG (H+L) | Invitrogen | Cat# A10042 | (1:1000) |
| Antibody | Alexa Fluor 647 donkey anti-mouse IgG (H+L) | Invitrogen | Cat# A31571 | (1:1000) |
| Antibody | Alexa Fluor 488 donkey anti-mouse IgG (H+L) | Invitrogen | Cat# A21202 | (1:1000) |
| Commercial assay or kit | ATP Assay Kit | Sigma-Aldrich | Cat# FLAA | |
| Commercial assay or kit | Adenosine Assay Kit | BioVision | Cat# K327-100 | |
| Chemical compound, drug | ATP-γ-S | Sigma-Aldrich | Cat# A1388; CAS: 93839-89-5 | |
| Chemical compound, drug | MesADP | Tocris | Cat# 1624; CAS: 475193-31-8 | |
| Chemical compound, drug | NECA | Sigma-Aldrich | Cat# E2387; CAS: 35920-39-9 | |
| Chemical compound, drug | CCPA | Sigma-Aldrich | Cat# C7938; CAS: 37739-05-2 | |
| Chemical compound, drug | CPT | Sigma-Aldrich | Cat# C102; CAS: 35873-49-5 | |
| Chemical compound, drug | SCH58261 | Tocris | Cat# 2270; CAS: 160098-96-4 | |
| Chemical compound, drug | ARL67156 trisodium salt hydrate | Sigma-Aldrich | Cat# A265; CAS: 160928-38-1 | |
| Chemical compound, drug | CGS 21680 hydrochloride | Tocris | Cat# 1063; CAS: 124431-80-7 | |
| Chemical compound, drug | Rhod-2 AM | Invitrogen | Cat# R1245MP | |
| Software, algorithm | GraphPad Prism (6.01) | GraphPad Software | https://www.graphpad.com/scientific-software/prism/ | |
| Software, algorithm | ImageJ | National Institutes of Health | https://imagej.nih.gov/ij/ | |

*Continued*

| Reagent type (species) or resource | Designation | Source or reference | Identifiers | Additional information |
|---|---|---|---|---|
| Software, algorithm | MATLAB R2017b | MathWorks | https://se.mathworks.com/products/matlab.html | |
| Software, algorithm | ANY-maze tracking software | Stoelting Co | https://www.stoeltingco.com/anymaze.html | |
| Software, algorithm | FREEZING (2.0.05) | Panlab Harvard Apparatus | https://www.panlab.com | |

## Animals

*GFAP-ChR2-EYFP* rats (Sprague-Dawley background) were generated at the Institute of Neuroscience, Chinese Academy of Sciences. To generate *GFAP-ChR2-EYFP* knock-in rats, we designed the single guide (sg) RNA near the stop codon in the last exon of the GFAP gene, and constructed a donor plasmid containing the ChR2-EYFP sequence. The plasmid was used as a template to repair the double-strand break by homologous recombination. Super-ovulated female Sprague-Dawley rats were mated to Sprague-Dawley males, and fertilized embryos were collected from the oviducts. Cas9 mRNA, sgRNAs, and donor were mixed and injected into the cytoplasm of fertilized eggs using a Narishige IM300 microinjector and the zygotes were cultured for several hours. Thereafter, 20–25 embryos were transferred into the oviducts of pseudopregnant Sprague-Dawley rats. The genotypes of mutant mice were determined by PCR of genomic DNA extracted from the tail.

Experiments were conducted on 2- to 4-month-old male and female *GFAP-ChR2-EYFP* rats and WT Sprague-Dawley rats. The rats were housed with food and water available ad libitum in a temperature-controlled room with a 12 hr light/dark cycle (lights on at 07:00). Rats were singly housed after surgery. All experimental procedures were approved by the Animal Advisory Committee at Zhejiang University (2019–2#) and were performed in strict accordance with the National Institutes of Health Guidelines for the Care and Use of Laboratory Animals (2006–398#). All surgeries were performed under sodium pentobarbital anesthesia, and every effort was made to minimize suffering.

## Stereotactic surgery for optical fiber and cannula implantation and in vivo optogenetic manipulation

Rats were deeply anesthetized with pentobarbital sodium (1%, wt/vol) and placed on a stereotaxic frame (Kopf, USA). Body temperature was kept stable throughout the procedure with a heating pad. A scalp incision was made with eye scissors. The skull was exposed and perforated with a stereotaxic drill at the target region. For optogenetic fiber implantation, two optical fibers (core diameter 200 μm, NA 0.37; Newdoon, China) were bilaterally implanted into CA1 (AP, –3.75 mm; ML,±2.46 mm; DV, –2.63 mm relative to bregma) for optogenetic manipulations. The optical fibers were connected to a laser source using a fiber optic patch cord (Newdoon, China). The intensity of laser stimulation was measured before each experiment at the tip of the optical fiber via a laser power meter (LP1, Sanwa, Japan). For pharmacological experiments, two guide cannulae (RWD Life Science, China) were bilaterally implanted into CA1 at the above coordinates for drug infusion. For photostimulation coupled with pharmacological experiments, an optical fiber (AP, –3.75 mm; ML, 1.85 mm; DV, –3.10 mm relative to bregma, at a 12° angle) and an infusion cannula (AP, –3.75 mm; ML, 3.01 mm; DV, –1.76 mm relative to bregma, at a 15° angle) were unilaterally implanted into CA1 for simultaneous photostimulation and drug infusion. The optical fibers and cannulae were fixed to the skull with dental cement. After the implantation surgery, rats were allowed to recover for 7 days before behavioral tests. After experiments, the positions of the optical fibers and cannulae were verified histologically. Rats with incorrect positioning of optical fibers or cannulae were excluded.

## In vivo virus microinjection

pZac2.1-GfaABC1D-mCherry-hPMCA2w/b was from Baljit Khakh Lab Plasmids in Addgene (Addgene plasmid # 111568). AAV5 GfaABC1D mCherry-hPMCA2w/b (7.0 × $10^{12}$ viral genomes/ml) and AAV5 GfaABC1D tdTomato (6.0 × $10^{12}$ viral genomes/ml) were constructed by Vigene Biosciences (Shandong, China). Rats were anesthetized with pentobarbital sodium (1%, wt/vol) and placed on a stereotaxic frame (Kopf, USA). Viruses were injected bilaterally into the dorsal CA1 (450 nl, 50 nl/

min) via a microsyringe. The needle was left in place for 15–20 min after the end of infusion to allow diffusion of the virus. After injection, mice were allowed 3 weeks of recovery and then performed behavioral tests.

## Fear-conditioning task

The fear-conditioning task was carried out in a 25 × 25 × 25 cm conditioning chamber (Panlab Harvard Apparatus, Spain) placed inside a sound-protected box. This task consisted of three phases: fear conditioning (training), testing for contextual fear (a hippocampus-dependent test) (*Kim and Fanselow, 1992*), and cued fear (a hippocampus-independent test) (*Phillips and LeDoux, 1992*). Rats were handled for 3 days before training was begun. On day 1, rats were placed into a fear conditioning chamber with a grid floor capable of delivering foot shocks, after the chamber was cleaned with 70% alcohol. The baseline freezing level was measured during a 2-min exploration period prior to the first conditioned stimulus (CS). Rats were then exposed to a 30 s tone (CS, 2 kHz, 85 dB) that co-terminated with a 2 s scrambled foot shock (unconditioned stimulus, US, 0.6 mA). A total of three tone-shock pairings were delivered with an inter-tone interval of 60 s. The rats remained in the conditioning chamber for a 90 s consolidation period following the last US. The conditioning phase lasted for 7 min, and all the processes were carried out in a relatively dark chamber. Twenty-four hours after conditioning, the rats were placed in the conditioning chamber for a 5-min contextual test. The cued test was carried out in a relatively bright chamber that had a context and smell different from the conditioning chamber (cleaned with 1% acetic acid). In the cued test, rats received a CS recall test (3 presentations of the 30 s CS alone with a 30-s inter-tone interval). The rats were considered to freeze if no movement was detected for 2 s. In the contextual test, the freezing level was calculated as the percentage of freezing time during 5 min in context. In fear-conditioning and the cued test, the freezing level was calculated as the percentage of freezing time during three presentations of the CS. The data were automatically recorded using commercial software (FREEZING, Panlab Harvard Apparatus, Spain).

When assessing the effect of astrocyte activation on fear memory, the rats received photostimulation for 15 min (473 nm, 10 Hz, 20 ms pulses, 1–3 mW at the fiber tip, 30 s light on, 30 s light off) immediately after fear conditioning or with an interval of 1, 2, and 3 hr. Sham operation was defined as surgery including the implantation of optical fibers and cannulae but without photostimulation. For pharmacological manipulations, drugs or vehicle were delivered by intracerebral or i.p. injection immediately after fear conditioning.

## Anxiety-like behavioral test

The OFT is a classical test to measure anxiety-like behavior in rodents (*Cai et al., 2018*). Rats were placed in the corner of a black open field arena (100 × 100 × 40 cm) at the start of the experiment and allowed to freely explore for 5 min. The center of the open field was defined as the central 50% of the arena. The locomotor activity of the rats in the open field was video-recorded and analyzed with automatic behavioral tracking software (ANY-maze, Stoelting Co., USA). The open field chamber was cleaned with 70% alcohol between animals. In the optogenetic studies, the total test time of 9 min was divided into three consecutive 3-min epochs consisting of stimulation off, stimulation on, and stimulation off periods (OFF-ON-OFF). The total distance was defined as the distance moved in a 3- or 5 min OFT. The center zone time was defined as the exploration time in the center area of the open field during a 3- or 5-min test.

## Object location recognition (OLR)

Rats were habituated to a square testing arena (100 cm × 100 cm × 40 cm) for 10 min per day for 3 consecutive days. After habituation, the OLR task was divided into a sample phase and a test phase, each lasting 5 min. In the sample phase, each rat was placed in the arena, exposed to two identical objects, and then returned to its home cage. After a 24 hr delay, each rat was returned to the arena for the test phase when it was exposed to the same objects as in the sample phase except that one of these objects (displaced object, DO) was moved to a novel location in the arena. The other (undisplaced object, UDO) remained in the original location. The time spent in exploring each object was recorded. Exploration was defined as touching the object with the nose or directing the nose to the object at a distance of no more than 2 cm. The percentage of time spent exploring an object was

defined as: (time at novel or familiar)/(time at novel + time at familiar). The discrimination index was calculated as: (time at novel − time at familiar)/(time at novel + time at familiar), where novel refers to the object in a novel location, and familiar refers to the other object.

## Microdialysis

Microdialysis was conducted on awake, freely moving rats as previously described (*Nam et al., 2016*) with modifications. For optical stimulation coupled with simultaneous microdialysis, adult rats were implanted stereotaxically with an optical fiber (AP, –3.75 mm; ML, 1.85 mm; DV, –3.10 mm relative to bregma, at a 12° angle) and a microdialysis probe guide cannula (AP, –3.75 mm; ML, 3.01 mm; DV, –1.76 mm relative to bregma, at a 15° angle; CMA 12, CMA Microdialysis AB, Sweden). Microdialysis sampling was started after 6 days of recovery from surgery. On the day of sampling, a microdialysis probe (CMA 12 Elite, membrane length 2 mm, CMA Microdialysis AB, Sweden) was inserted into CA1 through the probe cannula 3 hr before the start of microdialysis. The probe was connected to a syringe pump (CMA 402) with polyethylene tubing and perfused continuously with artificial cerebrospinal fluid (aCSF) at a constant flow rate of 1.5 µl/min. After pre-washing (30 min) and recovery (1 hr), 40 min samples of baseline dialysate were collected by an 820 microsampler (Univentor, Malta) as the first microdialysis. Then the second microdialysis for 40 min began with 15 min photostimulation. Extracellular fluid was collected in plastic vials in the presence of the ectonuclease inhibitor ARL67156 (100 µM final concentration). All the dialysis samples were stored at –80°C for later analysis.

## Measurement of extracellular ATP and adenosine

A previously described procedure for ATP measurement (*Zhang et al., 2003*) with some modifications was used. In brief, the extracellular ATP concentration in the samples was quantified with a bioluminescent assay kit (FLAA, Sigma-Aldrich). A calibration curve was generated with standard ATP samples and the luminescence of the dialysis medium was measured as the background ATP level. A 50 µl sample was added to 50 µl of ATP assay mix containing luciferase-luciferin buffer. The luminescence was measured by a luminometer (Varioskan Flash, Thermo Scientific, USA) according to the manufacturer's instructions.

The concentration of extracellular adenosine in the samples was quantified with an adenosine assay kit (Fluorometric, K327-100, BioVision) according to the protocol. Adenosine was measured using adenosine deaminase followed by a multi-step enzymatic approach resulting in the generation of an intermediate that reacts with the adenosine probe to form a fluorescent product. The fluorescent product was measured at excitation/emission = 535/587 nm.

## Drug administration

ATP-γ-S, NECA, CCPA, CPT and ARL67156 trisodium salt hydrate were from Sigma-Aldrich; MesADP, SCH58261 and CGS 21680 hydrochloride were from Tocris. NECA, CCPA, CPT, SCH58261, and CGS 21680 were made up to stock solution in dimethyl sulfoxide (DMSO) and then diluted to their final concentrations in sterile 0.9% saline. ATP-γ-S, MesADP, and ARL67156 were dissolved in sterile 0.9% saline and diluted to their final concentrations in sterile 0.9% saline. Intracerebral drug delivery was through previously-implanted infusion cannulae. On the day of the experiment, the internal cannulae that protruded 2 mm beyond the ends of the guide cannulae were inserted, and drugs (1 µl/side) were infused bilaterally. The vehicle control groups were given an equivalent amount of DMSO dissolved in sterile 0.9% saline or equivalent sterile 0.9% saline. For i.p. injection, rats were given CCPA (0.03, 0.1, 0.3, or 1 mg/kg body weight) or the appropriate vehicle.

## Immunohistochemistry and imaging

All rats were anesthetized with sodium pentobarbital and then perfused transcardially with 0.9% NaCl followed by 4% paraformaldehyde (PFA, wt/vol) dissolved in phosphate-buffered saline (PBS, pH 7.4). The brains were removed and postfixed in 4% PFA at 4°C overnight, then cryoprotected in 30% sucrose (wt/vol) for 3–4 days at 4°C. Coronal sections (40 µm) were cut on a microtome (CM 1950, Leica, Germany) and stored in PBS at 4°C for further use. For immunostaining, each section was treated with 0.5% Triton X-100 (vol/vol) for 10 min. After washing with PBS, the sections were blocked in 10% bovine serum albumin (BSA, wt/vol) with 5% donkey serum (wt/vol) for 1.5 hr at

room temperature and then incubated with primary antibody (rabbit anti-GFAP 1:500, Millipore; mouse anti-NeuN, 1:400, Millipore; Rabbit anti-DsRed 1:500, Clontech) diluted in 5% BSA (wt/vol) at 4°C for 24 hr. The sections were then washed three times (10 min each) in PBS, followed by incubation with secondary antibody (1:1000 Alexa Fluor 568 anti-rabbit, Invitrogen; 1:1000 Alexa Fluor 647 anti-mouse, Invitrogen; 1:1000 Alexa Fluor 488 donkey anti-mouse, Invitrogen) for 2 hr at room temperature. They were then incubated for 5 min with DAPI and rinsed three times with PBS. Finally, the sections were mounted on microscope slides and coverslipped. Fluorescence images were acquired with an Olympus FV-1200 (Japan) confocal microscope.

## Acute brain slice preparation

The acute brain slices were prepared following a previously described protocol (*Ting et al., 2018*). Briefly, *GFAP-ChR2-EYFP* rats (3–4 weeks postnatal) were anesthetized with pentobarbital sodium, then perfused transcardially with cold (2–4°C) oxygenated (95% O2/5% CO2) N-methyl-D-glucamine (NMDG) - 4-(2-hydroxyethyl)−1-piperazineethanesulfonic acid (HEPES) aCSF before decapitation. Then the whole brain was removed rapidly into cold oxygenated aCSF containing (in mM): 92 NMDG, 2.5 KCl, 1.25 $NaH_2PO_4$, 30 $NaHCO_3$, 20 HEPES, 25 glucose, 2 thiourea, 5 Na-ascorbate, 3 Na-pyruvate, 0.5 $CaCl_2 \cdot 4H_2O$, and 10 $MgSO_4 \cdot 7H_2O$. The pH of the NMDG–HEPES aCSF was titrated to pH 7.3–7.4 with concentrated HCl.

After the brain was swiftly dissected, transverse slices (300 μm) were cut on a vibratome (VT1200s, Leica, Germany) and transferred into a recovery chamber with NMDG-HEPES aCSF. Then, the slices were transferred into HEPES-aCSF (in mM): 92 NaCl, 2.5 KCl, 1.25 $NaH_2PO_4$, 30 $NaHCO_3$, 20 HEPES, 25 glucose, 2 thiourea, 5 Na-ascorbate, 3 Na-pyruvate, 2 $CaCl_2 \cdot 2H_2O$, and 2 $MgSO_4 \cdot 7H_2O$ (pH 7.3–7.4) at room temperature and allowed to recover for at least 1 hr. The recording aCSF contained (in mM): 119 NaCl, 2.5 KCl, 1.25 $NaH_2PO_4$, 24 $NaHCO_3$, 12.5 glucose, 2 $CaCl_2 \cdot 4H_2O$, and 2 $MgSO_4 \cdot 7H_2O$ (pH 7.3–7.4) continuously bubbled with 95% $O_2$/5% $CO_2$. For recording, an individual slice was transferred to a submerged recording chamber and continuously perfused with the above aCSF (3.0 ml/min) at 26°C. The slice was visualized under a microscope (BX51WI, Olympus, Japan) using infrared differential interference contrast optics.

## Electrophysiology

For whole-cell patch clamp recording from hippocampal slices, the patch electrodes were made from borosilicate glass capillaries (B-120-69-15, Sutter Instruments, USA) and had resistances in the 3–5 MΩ range. The internal solution contained (in mM): 130 K-gluconate, 4 KCl, 10 HEPES, 4 MgATP, 0.3 $Na_2GTP$, 10 $Na_2$-phosphocreatine, and 0.3 EGTA. Recordings were made with an Axon 700B patch-clamp amplifier and 1320A interface (Axon Instruments, USA). The signals were filtered at 2 kHz using amplifier circuitry, sampled at 10 kHz, and analyzed using Clampex 9.0 (Axon Instruments). Photostimulation was delivered by 473 nm solid-state laser diodes, and light pulses were generated with a custom-built high-speed shutter; the power density of the blue light was 1–3 mW/$mm^2$. Blue light was delivered to the slices through a thin quartz fiber (200 μm diameter, custom made).

## Calcium fluorescence imaging

$Ca^{2+}$ imaging in hippocampal slices was performed using a confocal laser scanning microscope (Olympus FV-1200, Japan). Astrocytes were bulk loaded in slices with Rhod-2 AM (20 μM, Invitrogen). The fluorescence intensity was measured at an excitation wavelength of 550 nm and emission wavelength of 580 nm. $Ca^{2+}$ signals were calculated as the relative change in fluorescence (ΔF/F), where F is the fluorescence intensity before photostimulation and ΔF is the change in fluorescence after photostimulation.

## Cell culture

Primary hippocampal cultures were prepared as described previously (*Kaech and Banker, 2006*) with some modifications. Embryonic day 18 rat hippocampi were dissected and dissociated with 0.125% trypsin. Cells were re-suspended in Dulbecco's modified Eagle's medium containing 10% fetal bovine serum and 10% Ham's F-12 (all from Gibco) at a cell density of 20,000 $mL^{-1}$. The neurons were plated on a layer of astrocytes and maintained at 37°C in a 5% $CO_2$ incubator. The culture

medium was changed every 2–3 days. The co-cultured cells were used for fluorescence image acquisition.

## Statistical analysis

All statistics were calculated with GraphPad Prism (Version 6.01). One-way or two-way repeated measures ANOVA followed by the Bonferroni or Newman-Keuls post hoc test and standard two-tailed paired or unpaired $t$-tests were used as indicated in the figure legends. Normal distribution was determined by D'Agostino-Pearson, Shapiro-Wilk, and Kolmogorov-Smirnov normality tests. Animals were randomly assigned to treatment groups. Data are presented as the mean ± SEM. Statistical significance was set at $p < 0.05$.

## Acknowledgements

We thank Hui-Fang Lou and Li-Ya Zhu for technical support. This work was supported by the National Key Research and Development Program (2016YFC1306700 and 2016YFA0501000), the National Natural Science Foundation of China (31970939, 81527901, 81821091, 31771167, and 31571090), the Non-profit Central Research Institute Fund of the Chinese Academy of Medical Sciences (2018PT31041), CAMS Innovation Fund for Medical Sciences (2019-I2M-5-057), Science and Technology Planning Project of Guangdong Province (2018B030331001), and Fundamental Research Funds for the Central Universities (2019FZA7009).

## Additional information

### Funding

| Funder | Grant reference number | Author |
| --- | --- | --- |
| National Key Research and Development Program | 2016YFC1306700 | Yan-Qin Yu |
| National Key Research and Development Program | 2016YFA0501000 | Shumin Duan |
| National Natural Science Foundation of China | 31970939 | Yan-Qin Yu |
| National Natural Science Foundation of China | 81527901 | Shumin Duan |
| Chinese Academy of Medical Sciences | Non-profit Central Research Institute Fund 2018PT31041 | Shumin Duan |
| Science and Technology Planning Project of Guangdong Province | 2018B030331001 | Shumin Duan Yan-Qin Yu |
| Fundamental Research Funds for the Central Universities | 2019FZA7009 | Shumin Duan |
| National Natural Science Foundation of China | 31771167 | Yan-Qin Yu |
| National Natural Science Foundation of China | 31571090 | Yan-Qin Yu |
| National Natural Science Foundation of China | 81821091 | Shumin Duan |
| CAMS Innovation Fund for Medical Science | 2019-I2M-5-057 | Shumin Duan |

The funders had no role in study design, data collection and interpretation, or the decision to submit the work for publication.

### Author contributions

Yulan Li, Conceptualization, Data curation, Formal analysis, Investigation, Writing - original draft, Writing - review and editing, Conceived and designed the experiments, Performed most of the

experiments including immunohistochemistry, optogenetics, behavioral tests, neuropharmacology, and microdialysis, Collected and analyzed the data, Wrote the manuscript; Lixuan Li, Data curation, Formal analysis, Investigation, Writing - original draft, Writing - review and editing, Designed the experiments, Performed most of the experiments including immunohistochemistry, optogenetics, behavioral tests, neuropharmacology, and microdialysis, Collected and analyzed the data, Wrote the manuscript; Jintao Wu, Data curation, Performed electrophysiology experiments; Zhenggang Zhu, Xiang Feng, Liming Qin, Data curation, Performed cell culture experiments; Yuwei Zhu, Data curation, Helped with transcardial perfusion in the revision stage; Li Sun, Yijun Liu, Formal analysis; Zilong Qiu, Resources, Helped in generating the GFAP-ChR2-EYFP rats; Shumin Duan, Yan-Qin Yu, Conceptualization, Supervision, Funding acquisition, Writing - original draft, Writing - review and editing

### Author ORCIDs
Yan-Qin Yu (iD) https://orcid.org/0000-0002-4378-5931

### Ethics
Animal experimentation: All experimental procedures were approved by the Animal Advisory Committee at Zhejiang University (2019-2#) and were performed in strict accordance with the National Institutes of Health Guidelines for the Care and Use of Laboratory Animals (2006-398#). All surgery was performed under sodium pentobarbital anesthesia, and every effort was made to minimize suffering.

### Decision letter and Author response
Decision letter https://doi.org/10.7554/eLife.57155.sa1
Author response https://doi.org/10.7554/eLife.57155.sa2

## Additional files

### Supplementary files
• Transparent reporting form

### Data availability
All data generated or analysed during this study are included in the manuscript and supporting files. Source data files have been provided for all manuscript figures. Source data has been provided online at https://doi.org/10.5061/dryad.p8cz8w9mc.

The following dataset was generated:

| Author(s) | Year | Dataset title | Dataset URL | Database and Identifier |
|---|---|---|---|---|
| Li Y, Li L, Wu J, Zhu Z, Feng X, Qin L, Zhu Y, Sun L, Liu Y, Qiu Z, Duan S, Yu YQ | 2020 | Activation of Astrocytes in Hippocampus Decreases Fear Memory through Adenosine A1 Receptors | https://doi.org/10.5061/dryad.p8cz8w9mc | Dryad Digital Repository, 10.5061/dryad.p8cz8w9mc |

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
