## [Decision Letter]

**Acceptance summary:**

Your work further highlights the important role of glial cells in the tripartite synapse and provides potential targets for treatment of disorders associated with excessive anxiety.

**Decision letter after peer review:**

Thank you for submitting your article "Activation of astrocytes in hippocampus decreases fear memory through adenosine A_1_ receptors" for consideration by *eLife*. Your article has been reviewed by three peer reviewers including Margaret M McCarthy as the Reviewing Editor and Reviewer #1, and the evaluation has been overseen by Laura Colgin as the Senior Editor. The following individuals involved in review of your submission have agreed to reveal their identity: Long-Jun Wu (Reviewer #2).

The reviewers have discussed the reviews with one another and the Reviewing Editor has drafted this decision to help you prepare a revised submission.

Summary:

Li et al., present a well-written, well-organized and potentially impactful study on the role of astrocyte signaling in fear memory using optogenetic manipulation specifically in astrocytes. The authors generated GFAP-ChR2-EYFP rats to enable optogenetic manipulation of astrocytes in rats. They found that optogenetic activation of astrocytes impaired memory consolidation. Pharmacological interrogations revealed the critical role of adenosine and A_1_Rs in the fear memory impairment by astrocyte activation. Overall, the study is very well-designed and thorough. Multiple approaches were used and the results are convincing. However, there are some concerns. The most important of which is the lack of evidence for a role for astrocytes in fear learning under normal conditions. See below for further discussion of this point.

Essential revisions:

1) Ultimately the conclusion that astrocytes regulate fear memory is not sufficiently established. What is missing is a definitive connection between astrocyte activation in fear conditioning and extinction under normal conditions. It is well established that adenosine lowers arousal. Also, that arousal affects performance in learning tests. Regardless of how adenosine is administered, these facts will obtain. Adenosine increased by optogenetic stimulation will necessarily have the same effect as drug administration to alter adenosine levels. In that sense, these experiments are a demonstration of established facts, rather than an experimental test of the hypothesis that astrocytes control fear learning. The missing link is whether and how astrocytes would be activated during fear conditioning to alter learning through adenosine release. In order to reach the conclusions proposed, experimental evidence that astrocytes are required in the normal behavior (fear memory) by acting through adenosine, is required to complement the present findings that artificial stimulation of the cells impairs function.

2) Microdialysis experiments indicated an increase in adenosine from ~3 nM to ~3.5 nM following photostimulation. This increase seems very underwhelming given the magnitude of behavioral changes ascribed to it. The author should discuss (based on A_1_R pharmacokinetics) how this slight increase in interstitial adenosine could have such a profound effect on the molecular and synaptic underpinnings of memory. Along the similar line, the specific time window (within one hour after fear conditioning) of effective photostimulation on behaviors is quite amazing, but needs more explanation.

3) The authors investigated the effects of ATP and ADP by infusing stable forms of these compounds (ATP-γ-S or MeSADP) into the hippocampus. The authors did not find that introduction of either compound in the mM range impacts their core mechanism of adenosine signaling to the A_1_ receptor. However, both of these compounds in the mM range should certainly evoke large astrocyte calcium activity through Gq-coupled P2Y or ionotropic P2X receptors expressed on astrocytes. The authors showed that their ChR2 activation increases astrocyte calcium activity (Figure 1), and this increase in calcium (potentially through non-selective cation flux through ChR2) is presumably upstream of increased ATP release and adenosine levels. So why does the potential for high astrocyte calcium activity with local ATP-γ-S or MeSADP infusion not feed into the upstream mechanism for fear memory impairment?

4) Only hippocampus is stimulated. One would like to see control experiments in which other brain regions that are not involved in the learning task are stimulated to release adenosine. The results would add information on the specificity being argued that astrocytes in CA1 regulate fear conditioning, rather than affecting performance by altering general arousal levels in the animal.

5) The authors argue their findings are most relevant to PTSD which is a disorder that occurs 2-4 times more frequently in women. Yet, all of the experiments are conducted on male rats. Thus, the conclusions reached at this time are relevant only to men and therefore half the overall population and far less of the clinically relevant population. Throughout the manuscript conclusions should be changed so that instead of referring to "rats", the statement "male rats" is made. Arguments about potential effects of the estrus cycle are baseless in this circumstance.

6) One of the major technical innovations in this study is to use the newly generated GFAP-ChR2-EYFP rats. More information should be provided on how the transgenic rats were generated in Materials and methods section. In addition, characterization of ChR2 expression in different brain areas other than hippocampus will be informative, considering the recent concern of GFAP as astrocyte specific promoter. Even in the hippocampus, the newly born neurons in the dentate gyrus may also transit through a period of GFAP positivity.

7) Is it possible that non-physiological effects of ChR2 on astrocyte function (such as membrane stretch, or even cell death) could be leading to the increased ATP/adenosine increases? If such increases are a non-physiological artifact of the system used, the interpretation of the present results will be totally different. These points should be discussed intensively.

8) Figure 3D-G, why did the rats with retraining at D27 have very low freezing baseline? They were shown to have contextual fear memory which still has 30-40% freezing at D26 (Figure 3E). Instead of fear retraining, another form of hippocampus-dependent memory test might be more relevant to draw the conclusion of "astrocyte activation does not affect new learning".

9) We suggest that the title be changed to "Activation of astrocytes in male rat hippocampus decreases fear memory through adenosine A_1_ receptors".

---

## [Author Response]

Summary:Li et al., present a well-written, well-organized and potentially impactful study on the role of astrocyte signaling in fear memory using optogenetic manipulation specifically in astrocytes. The authors generated GFAP-ChR2-EYFP rats to enable optogenetic manipulation of astrocytes in rats. They found that optogenetic activation of astrocytes impaired memory consolidation. Pharmacological interrogations revealed the critical role of adenosine and A_1_Rs in the fear memory impairment by astrocyte activation. Overall, the study is very well-designed and thorough. Multiple approaches were used and the results are convincing. However, there are some concerns. The most important of which is the lack of evidence for a role for astrocytes in fear learning under normal conditions. See below for further discussion of this point.Essential revisions:1) Ultimately the conclusion that astrocytes regulate fear memory is not sufficiently established. What is missing is a definitive connection between astrocyte activation in fear conditioning and extinction under normal conditions. It is well established that adenosine lowers arousal. Also, that arousal affects performance in learning tests. Regardless of how adenosine is administered, these facts will obtain. Adenosine increased by optogenetic stimulation will necessarily have the same effect as drug administration to alter adenosine levels. In that sense, these experiments are a demonstration of established facts, rather than an experimental test of the hypothesis that astrocytes control fear learning. The missing link is whether and how astrocytes would be activated during fear conditioning to alter learning through adenosine release. In order to reach the conclusions proposed, experimental evidence that astrocytes are required in the normal behavior (fear memory) by acting through adenosine, is required to complement the present findings that artificial stimulation of the cells impairs function.

We thank the reviewer for this valuable suggestion. We agree with the reviewer that adenosine lowers arousal. However, our major conclusion of photostimulation- and drug-induced fear memory attenuation is supported by evidence including: (a) we delivered the photostimulation and drugs after but not before or during the Pavlovian fear conditioning. Our protocol aimed to affect the memory consolidation process rather than the learning process, and indeed the learning curves of control and treated rats were similar, showing that they had similar performance in learning (Figure 2, Figure 4, Figure 5 and Figure 6); and (b) astrocyte activation by photostimulation in hippocampal CA1 specifically decreased the hippocampus-dependent contextual fear memory but not the hippocampus-independent cued fear memory (Figure 2). Arousal changes the general state of rats. By affecting arousal, the animal would show a general variation in fear memory without selectivity for contextual or cued fear memory. These results showed that astrocyte activation specifically decreased contextual fear memory through the disruption of memory consolidation, suggesting that these cannot be the result of changes in arousal.

Following the reviewer's valuable suggestion, we further tested whether astrocyte activity is required for fear memory attenuation. Astrocyte Ca^2+^ signaling was attenuated by the expression of the Ca^2+^-extruder PMCA2w/b (plasma membrane Ca^2+^ ATPase isoform 2 splice variant w/b) in hippocampal astrocytes (Yu et al., 2020; Yu et al., 2018). So, we used an adeno-associated virus (AAV5) expressing human PMCA2w/b (hPMCA2w/b) with an astrocyte-specific GfaABC1D promoter to reduce hippocampal astrocyte Ca^2+^ signaling (Figure 2G and H). In the control group, we microinjected tdTomato instead of hPMCA2w/b (Figure 2—figure supplement 4A and B). We ran behavioral tests 21 days after viral injection and found that control and hPMCA2w/b rats did not differ in distance moved and center zone exploration time in the open field test (Figure 2—figure supplement 4C and D). To assess whether fear memory is affected by reducing hippocampal astrocyte Ca^2+^ signaling, we used the Pavlovian fear conditioning paradigm again, and the contextual and cued fear memory were measured on day 2 after fear acquisition. We found that control and hPMCA2w/b rats had comparable learning curves for fear conditioning (Figure2—figure supplement 4E). Interestingly, contextual fear memory but not cued fear memory (Figure 2J and K) was significantly increased in hPMCA2w/b rats. These results demonstrated that hippocampal astrocyte activity is indeed required for contextual fear memory.

2) Microdialysis experiments indicated an increase in adenosine from ~3 nM to ~3.5 nM following photostimulation. This increase seems very underwhelming given the magnitude of behavioral changes ascribed to it. The author should discuss (based on A_1_R pharmacokinetics) how this slight increase in interstitial adenosine could have such a profound effect on the molecular and synaptic underpinnings of memory. Along the similar line, the specific time window (within one hour after fear conditioning) of effective photostimulation on behaviors is quite amazing, but needs more explanation.

We thank the reviewer for raising this important concern.

Our explanation is:

a) Microdialysis experiments showed an increase of adenosine (first, 2.88 ± 0.17 μM, ^second^, 3.42 ± 0.11 μM) following photostimulation (Figure 4). The increase is statistically significant and is ~0.5 μM in absolute value. And the real extracellular adenosine concentration is higher than that in the dialysate because adenosine is metabolized extremely fast, and in the process of sample collection, the material in the probe and tubing, flow rate, probe length, and recovery can influence the quantitative estimation of extracellular fluid concentrations (Hammarlund-Udenaes, 2017). So, the actual increase in absolute adenosine concentration may be higher in vivo.

b) The adenosine A_1_ receptor (A_1_R) is a G-protein-coupled receptor (GPCR) that preferentially couples to inhibitory Gi/o heterotrimeric G proteins. In the pharmacology of the GPCR-ligand complex, a very small increase in the fractional receptor occupancy can cause a dramatic increase in the tissue response as shown in the most common hyperbolic relationship between the receptor occupancy of acetylcholine in guinea-pig ileum and the tissue response (Kenakin, 2017). These data indicate that when the adenosine level rises to reach a certain A_1_R occupancy threshold in vivo, a profound effect on memory can occur. We have discussed how adenosine and A_1_Rs influence the neuronal excitability and memory in the Discussion section in the revised manuscript.

c) We followed the reviewer's suggestion and provided further explanation and references for the specific time window in the Discussion section in the revised manuscript: "Memory consolidation is a molecular and cellular processes by which newly-acquired information is gradually stabilized by strengthening synaptic connections after the initial training (Dudai, 2004). It has been proposed that long-term potentiation (LTP), a form of synaptic plasticity, is an intrinsic property of this consolidation (Abraham and Williams, 2008; Kandel et al., 2014). Once the LTP or memory has passed into the protein synthesis-dependent phase, LTP is highly resistant to disruption, while memory becomes more difficult to erase and is said to have been consolidated (Abraham and Williams, 2008; Mednick et al., 2011). So if the time between learning and interference is sufficiently long for the processes associated with LTP to have occurred, the memory trace would become less vulnerable. Our present study showed that photoactivation of astrocytes in CA1 within the early memory consolidation phase (1 hour) but not beyond it (2 hours, 3 hours or during retrieval) after a fear stimulus produced a long-lasting attenuation of contextual fear memory and an anxiolytic effect. This finding is in accordance with reports that new memories are sensitive to interference within a short time (1 hour) after learning, but not after a relatively long interval (6 hours) during which the memory trace becomes consolidated (Mednick et al., 2011)."

3) The authors investigated the effects of ATP and ADP by infusing stable forms of these compounds (ATP-γ-S or MeSADP) into the hippocampus. The authors did not find that introduction of either compound in the mM range impacts their core mechanism of adenosine signaling to the A_1_ receptor. However, both of these compounds in the mM range should certainly evoke large astrocyte calcium activity through Gq-coupled P2Y or ionotropic P2X receptors expressed on astrocytes. The authors showed that their ChR2 activation increases astrocyte calcium activity (Figure 1), and this increase in calcium (potentially through non-selective cation flux through ChR2) is presumably upstream of increased ATP release and adenosine levels. So why does the potential for high astrocyte calcium activity with local ATP-γ-S or MeSADP infusion not feed into the upstream mechanism for fear memory impairment?

We appreciate the reviewer's comments.

Our explanation is:

a) P2X and P2Y receptors are widely distributed throughout the brain, in both neurons and glia. ATP-γ-S and MesADP induce a [Ca^2+^]_i_ increase in many types of neurons and glia. These signals are naturally complex and involve both P2Y-mediated Ca^2+^ release from the intracellular stores and P2X-mediated Ca^2+^ influx (Burnstock et al., 2011; Lalo et al., 1998). It is likely that the potential effect of astrocyte Ca^2+^ activity with local ATP-γ-S or MesADP infusion on fear memory was obscured by their effects on neurons.

b) Specific ChR2 expression induces Ca^2+^ elevation in astrocytes and has been used as a tool for astrocyte activation (Chen et al., 2013; Figueiredo et al., 2011; Gourine et al., 2010; Tan et al., 2017). Our previous work showed that the ChR2-induced [Ca^2+^]_i_ elevation in astrocytes is entirely due to Ca^2+^ influx from the extracellular space, and this is mostly attributed to the Na^+^-Ca^2+^ exchanger following Na^+^ influx through ChR2 channels (Yang et al., 2015). Unlike the activation of metabotropic receptors by local ATP-γ-S or MesADP, the ChR2-induced [Ca^2+^]_i_ elevation is the result of membrane channel-mediated Ca^2+^ entry from the extracellular space, but not from intracellular ER Ca^2+^ release. Due to the proximity to the plasma membrane where exocytosis occurs, transmembrane Ca^2+^ influx may be more efficient in inducing gliotransmitter release than Ca^2+^ release from intracellular stores (Chen et al., 2013; Tan et al., 2017; Yang et al., 2015). Some description has been added into the Introduction in the revised manuscript.

c) In addition, previous studies have demonstrated that different stimulation patterns may induce astrocytes to release different substances that then activate their corresponding receptors on nearby neurons, leading to different forms of synaptic regulation (Covelo and Araque, 2018). Therefore, a local ATP-γ-S- and MeSADP-induced increase of [Ca^2+^]_i_ did not affect fear memory probably due to their different stimulation patterns compared to photostimulation.

4) Only hippocampus is stimulated. One would like to see control experiments in which other brain regions that are not involved in the learning task are stimulated to release adenosine. The results would add information on the specificity being argued that astrocytes in CA1 regulate fear conditioning, rather than affecting performance by altering general arousal levels in the animal.

As we described in comment 1, our major conclusion that photostimulation- induced fear memory attenuation is not induced by altering general arousal levels.

We followed the reviewer's suggestion and performed photostimulation in the motor cortex (AP, –0.5 mm; ML, ±1.6 mm; DV, –0.9 mm relative to bregma) after the Pavlovian fear conditioning paradigm. The motor cortex is responsible for planning and executing actions, motor learning, and coordination (Li et al., 2015; Svoboda and Li, 2018). We found that the photostimulated rats showed no difference in contextual fear memory compared with controls (Author response image 1).

**Author response image 1. sa2fig1:** Photostimulation in the motor cortex of *GFAP-ChR2-EYFP* rats after fear conditioning has no effect on contextual fear memory. (A) Schematic of the experimental design for fear conditioning, photostimulation, and the subsequent test protocols. (B) Freezing levels of control (sham operation, n = 7) and photostimulated *GFAP-ChR2-EYFP* rats (n = 8) during fear conditioning. (C) Freezing levels tested on day 2 of control and photostimulated rats (p = 0.6548, Student’s unpaired two-tailed t-test).

5) The authors argue their findings are most relevant to PTSD which is a disorder that occurs 2-4 times more frequently in women. Yet, all of the experiments are conducted on male rats. Thus, the conclusions reached at this time are relevant only to men and therefore half the overall population and far less of the clinically relevant population. Throughout the manuscript conclusions should be changed so that instead of referring to "rats", the statement "male rats" is made. Arguments about potential effects of the estrus cycle are baseless in this circumstance.

We thank the reviewer for these comments. Epidemiological investigations have shown that females are twice as likely to develop PTSD as males (Breslau, 2001; Parsons and Ressler, 2013). However, the chance of exposure to traumatic events such as military combat, disaster, and criminal violence is higher in males than females (Breslau, 2001; Parsons and Ressler, 2013). In populations such as war veterans, police officers, and firefighters who work in highly exposed traumatic situations, there are more males. The Pavlovian fear conditioning paradigm is a commonly-used animal model of PTSD (Mahan and Ressler, 2012). Many studies have used male mice or rats to study the cellular mechanisms and circuitry of fear memory for the potential therapy of PTSD (Han et al., 2009; Khalaf et al., 2018; Rossetti et al., 2017; Zhang et al., 2020).

Incorporating both sexes would aid to more comprehensively understand and infer links with clinical symptoms (McCarthy et al., 2017). Following the reviewer's suggestion, we examined the effect of astrocyte activation on fear memory in female rats. Photostimulation in the dorsal CA1 followed the Pavlovian fear conditioning paradigm, and contextual and cued fear memory were measured on day 2 after fear acquisition (Figure 2—figure supplement 1A). Consistent with the results in male rats, the photostimulated female rats showed no difference in freezing levels during fear conditioning compared with controls (sham operation) (Figure 2—figure supplement 1B). Contextual fear memory but not cued fear memory (Figure 2—figure supplement 1C and D) was significantly decreased in the photostimulated females. These results showed that the fear memory attenuation induced by astrocyte activation is present in both males and females. Some descriptions have been added to the Results section in the revised manuscript.

6) One of the major technical innovations in this study is to use the newly generated GFAP-ChR2-EYFP rats. More information should be provided on how the transgenic rats were generated in Materials and methods section. In addition, characterization of ChR2 expression in different brain areas other than hippocampus will be informative, considering the recent concern of GFAP as astrocyte specific promoter. Even in the hippocampus, the newly born neurons in the dentate gyrus may also transit through a period of GFAP positivity.

We thank the reviewer for the valuable suggestions, following which we describe the generation of *GFAP-ChR2-EYFP* rats in the Materials and methods section of the revised manuscript: "To generate GFAP-ChR2-EYFP knock-in rats, we designed the single guide (sg) RNA near the stop codon in the last exon of the GFAP gene, and constructed a donor plasmid containing the ChR2-EYFP sequence. The plasmid was used as a template to repair the double-strand break by homologous recombination. Super-ovulated female Sprague-Dawley rats were mated to Sprague-Dawley males, and fertilized embryos were collected from the oviducts. Cas9 mRNA, sgRNAs, and donor were mixed and injected into the cytoplasm of fertilized eggs using a Narishige IM300 microinjector and the zygotes were cultured for several hours. Thereafter, 20–25 embryos were transferred into the oviducts of pseudopregnant Sprague-Dawley rats. The genotypes of mutant mice were determined by PCR of genomic DNA extracted from the tail."

In addition, we followed the reviewer's suggestion by performing IHC experiments and confirmed the specific expression of ChR2 in different brain areas such as the motor cortex, lateral posterior thalamic nucleus, and dorsomedial hypothalamic nucleus. We observed the co-localization of EYFP labeling with the specific astrocytic marker GFAP, but not with the neuronal marker NeuN. These data are summarized in Figure 1—figure supplement 1 in the revised manuscript.

7) Is it possible that non-physiological effects of ChR2 on astrocyte function (such as membrane stretch, or even cell death) could be leading to the increased ATP/adenosine increases? If such increases are a non-physiological artifact of the system used, the interpretation of the present results will be totally different. These points should be discussed intensively.

We thank the reviewer. It has been demonstrated that ChR2 expression is nontoxic, safe, stable, and functional (Aravanis et al., 2007; Cardin et al., 2010; Deisseroth, 2011; Doroudchi et al., 2011; Zhang et al., 2006). At present, ChR2 is widely used to explore the role of glia in regulating rodent behavior and circuits by precisely manipulating their Ca^2+^ signaling (Bang et al., 2016; Figueiredo et al., 2011; Gourine et al., 2010; Nam et al., 2016; Perea et al., 2014; Yamashita et al., 2014). We have added the above statement to the Introduction in the revised manuscript.

In the present study, we found that photostimulation of astrocytes triggered increased ATP and adenosine concentrations in CA1. The gliotransmitter adenosine and A_1_Rs were responsible for the fear memory attenuation and anxiolytic effect. Similarly, astrocytic ATP release has also been induced by optogenetic stimulation of ChR2-expressing spinal astrocytes and induces pain hypersensitivity through adenosine receptors (Nam et al., 2016).

8) Figure 3D-G, why did the rats with retraining at D27 have very low freezing baseline? They were shown to have contextual fear memory which still has 30-40% freezing at D26 (Figure 3E). Instead of fear retraining, another form of hippocampus-dependent memory test might be more relevant to draw the conclusion of "astrocyte activation does not affect new learning".

We apologize for the confusion. On day 27, the rats were retrained in a new conditioning chamber, so the baseline freezing level was ~10%. We have addressed this issue in the Results section in the revised manuscript.

Following the reviewer’s suggestion, we repeated the experiments in which the rats receiving photostimulation immediately after fear acquisition showed a persistent decrease of contextual fear memory lasting up to 26 days (Figure 3D and E). On day 27 the photostimulated rats (conditioned) performed the object location recognition (OLR) task and spent percentages of time exploring the object similar to controls in the sample phase (Figure 3H), indicating that astrocyte activation does not affect new learning. Furthermore, in the test phase of OLR on day 28, the conditioned rats spent more time exploring the object in a novel location relative to the object in a familiar location (Figure 3I) and the discrimination index was similar to controls (Figure 3J). These data suggest that astrocyte activation does not affect new memory formation. We describe the protocol of the OLR task in the Materials and methods section in the revised manuscript.

9) We suggest that the title be changed to "Activation of astrocytes in male rat hippocampus decreases fear memory through adenosine A_1_ receptors".

We thank the reviewer. As described in comment 5, we would like to keep the original title.